# Topaz-Denoise: general deep denoising models for cryoEM and cryoET

Tristan Bepler [1,2], Kotaro Kelley[3,5], Alex J. Noble [3✉] & Bonnie Berger [2,4✉]

Cryo-electron microscopy (cryoEM) is becoming the preferred method for resolving protein structures. Low signal-to-noise ratio (SNR) in cryoEM images reduces the confidence and throughput of structure determination during several steps of data processing, resulting in impediments such as missing particle orientations. Denoising cryoEM images can not only improve downstream analysis but also accelerate the time-consuming data collection process by allowing lower electron dose micrographs to be used for analysis. Here, we present Topaz-Denoise, a deep learning method for reliably and rapidly increasing the SNR of cryoEM images and cryoET tomograms. By training on a dataset composed of thousands of micrographs collected across a wide range of imaging conditions, we are able to learn models capturing the complexity of the cryoEM image formation process. The general model we present is able to denoise new datasets without additional training. Denoising with this model improves micrograph interpretability and allows us to solve 3D single particle structures of clustered protocadherin, an elongated particle with previously elusive views. We then show that low dose collection, enabled by Topaz-Denoise, improves downstream analysis in addition to reducing data collection time. We also present a general 3D denoising model for cryoET. Topaz-Denoise and pre-trained general models are now included in Topaz. We expect that Topaz-Denoise will be of broad utility to the cryoEM community for improving micrograph and tomogram interpretability and accelerating analysis.

[1] Computational and Systems Biology, MIT, Cambridge, MA, USA. [2] Computer Science and Artificial Intelligence Laboratory, MIT, Cambridge, MA, USA. [3] National Resource for Automated Molecular Microscopy, Simons Electron Microscopy Center, New York Structural Biology Center, New York, NY, USA. [4] Department of Mathematics, MIT, Cambridge, MA, USA. [5] Deceased: Kotaro Kelley. ✉email: anoble@nysbc.org; bab@mit.edu

Visualization of micrographs from cryo-electron microscopy (cryoEM) of biological specimens is primarily limited by the phase contrast of proteins, the low electron dose conditions required due to radiation damage accrued by the proteins, and the thickness of the ice. As researchers push towards smaller and smaller proteins, these issues hinder downstream analyses because these proteins become increasingly difficult to distinguish from noise. Certain orientations of large, non-globular proteins can also have low signal, leading to missing views. The typical signal-to-noise ratio (SNR) of a cryoEM micrograph is estimated to be only as high as 0.1[1], amongst the lowest in any imaging field, and no ground truth exists. Nonetheless, several steps during collection and processing of micrographs in single particle cryoEM rely on properly human-inspecting micrographs, identifying particles, and examining processed data. Conventional cryoEM methods for improving contrast in micrographs include downsampling, bandpass filtering, and Wiener filtering[2,3]. However, these methods do not address the specific noise properties of micrographs and often do not provide interpretable results, which increasingly hinders attempts to resolve small and non-globular proteins[4,5].

At the same time, there is a push in the field to fund large research facilities for high-throughput cryoEM. These and smaller facilities are moving towards the synchrotron model of data collection and need to increase their throughput to meet rising demand. One approach to speed up collection would be to collect shorter micrograph exposures. However, reducing total dose would exacerbate SNR-related analysis problems. Better micrograph denoising provides the opportunity to reduce total dose and increase collection throughput without compromising interpretability or downstream results.

Image denoising has long been a topic of significant interest in the computer vision and signal processing community[6], but has recently seen a surge in interest from the machine learning community. Advances in deep neural networks have enabled substantial improvements in image restoration and inpainting (i.e. filling in missing pixels) by learning complex, non-linear priors over the applied image domain. However, these methods require ground truth images to provide supervision for learning the denoising model[7,8], and are hence limited to domains where ground truth is available. To overcome this barrier, Lehtinen et al.[9] presented a general machine learning (ML) framework, called Noise2Noise, for learning denoising models from paired noisy images rather than paired noisy and ground truth images. This method has been followed by several others for learning denoising models without ground truth[10–12]. These methods offer new approaches for training deep neural network models for denoising in challenging domains. In cryoEM, neural network denoising software has only just started to emerge for dataset-by-dataset cryo-electron tomogram (cryoET) denoising[13,14] and single particle micrograph denoising[15]. However, there have not been any systematic evaluations of these methods to date nor have pre-trained general denoising models been developed.

Here, we develop Topaz-Denoise, large-scale, publicly available denoising models for cryoEM and cryoET. Conventional cryoEM and cryoET denoising methods are ad-hoc filters that do not model the complex image generative process. To address this, our goal is to learn the denoising process directly from data. However, deep denoising models typically require ground truth signal which is not available in cryoEM. We make the key insight that the individual movie frames collected by modern direct detector devices (DDD) are many independent observations of the same underlying signal and, hence, can be used to learn denoising models directly via the Noise2Noise framework. Trained on thousands of micrographs from DDD - K2, Falcon II, and Falcon III - across a variety of imaging conditions, these general models (also called pre-trained models) provide robust denoising without the need to train on a dataset-by-dataset basis. We test and compare these denoising models on several micrographs of typical particles and of small particles, study improvements in SNR, and use denoising combined with Topaz particle picking[16] to obtain 3D single particle cryoEM structures of clustered protocadherin, an elongated particle with previously-elusive views and a putative new conformation. We also show that denoising enables more rapid data collection by allowing micrographs to be collected with a lower electron total dose (10–25% typical exposure times) without sacrificing interpretability or downstream processing. Shorter exposure times allow for higher throughput microscope usage, which reduces research cost and increases research efficiency. In addition, we develop a general 3D denoising model for cryoET tomograms, trained on dozens of cryoET tomograms, and show that our general denoising model performs comparably to models trained on a dataset-by-dataset basis. These models are integrated into Topaz allowing easy access to the community, along with the denoising framework that allows users to train their own cryoEM and cryoET denoising models.

Topaz-Denoise source code is freely available as part of Topaz (http://topaz.csail.mit.edu) and can be installed through Anaconda, Pip, Docker, Singularity, and SBGrid[17], and is now integrated into CryoSPARC[18], Relion[19], Appion[20] and Scipion[21]. As with Topaz, Topaz-Denoise is designed to be modular and can easily be integrated into other cryoEM software suites. Topaz-Denoise includes several pre-trained models and the ability for the user to train their own models. Topaz-Denoise 2D training and inference runs efficiently on a single GPU computer, while 3D training and inference runs efficiently on multi-GPU systems. Both 2D and 3D denoising are integrated into the standalone Topaz GUI to assist with command generation.

## Results

**Denoising with Topaz improves micrograph interpretability and SNR.** We develop a general cryoEM micrograph denoising model by training a neural network using the Noise2Noise framework on dozens of representative datasets of commonly used imaging conditions (Fig. 1, "Methods"). By learning the denoising model directly from data, we avoid making specific assumptions about the noise-generating process leading to superior denoising performance.

Denoising with Topaz improves micrograph interpretability by eye on several datasets and improves SNR measurements in quantitative analyses. Our model correctly smoothes background areas while preserving structural features better than conventional methods (i.e. affine or low-pass filtering) (Fig. 1 and Supplementary Figs. 1–4). Given this known smoothing behavior of micrograph areas containing primarily noise, we find that denoising allows for the identification of structured background features from noise. Figure 1 shows two micrographs where the background areas between particles are flattened after denoising, while Supplementary Fig. 5 shows microtubules with known small proteins in background areas properly retained after denoising. Our denoising model has the combined advantage of reducing ambiguity as to whether the background of a micrograph is generally free from contamination, allowing researchers to identify small and/or low density particle views, for example as applied to micrographs from Mao et al.[22] (Supplementary Figs. 6 and 7). In these types of scenarios, visual assessment of denoised micrographs compared to raw micrographs increases protein density confidence, increases confidence of background content, and reduces eye strain for researchers.

We quantitatively assess denoising performance by measuring the SNR of raw micrographs, micrographs denoised with our

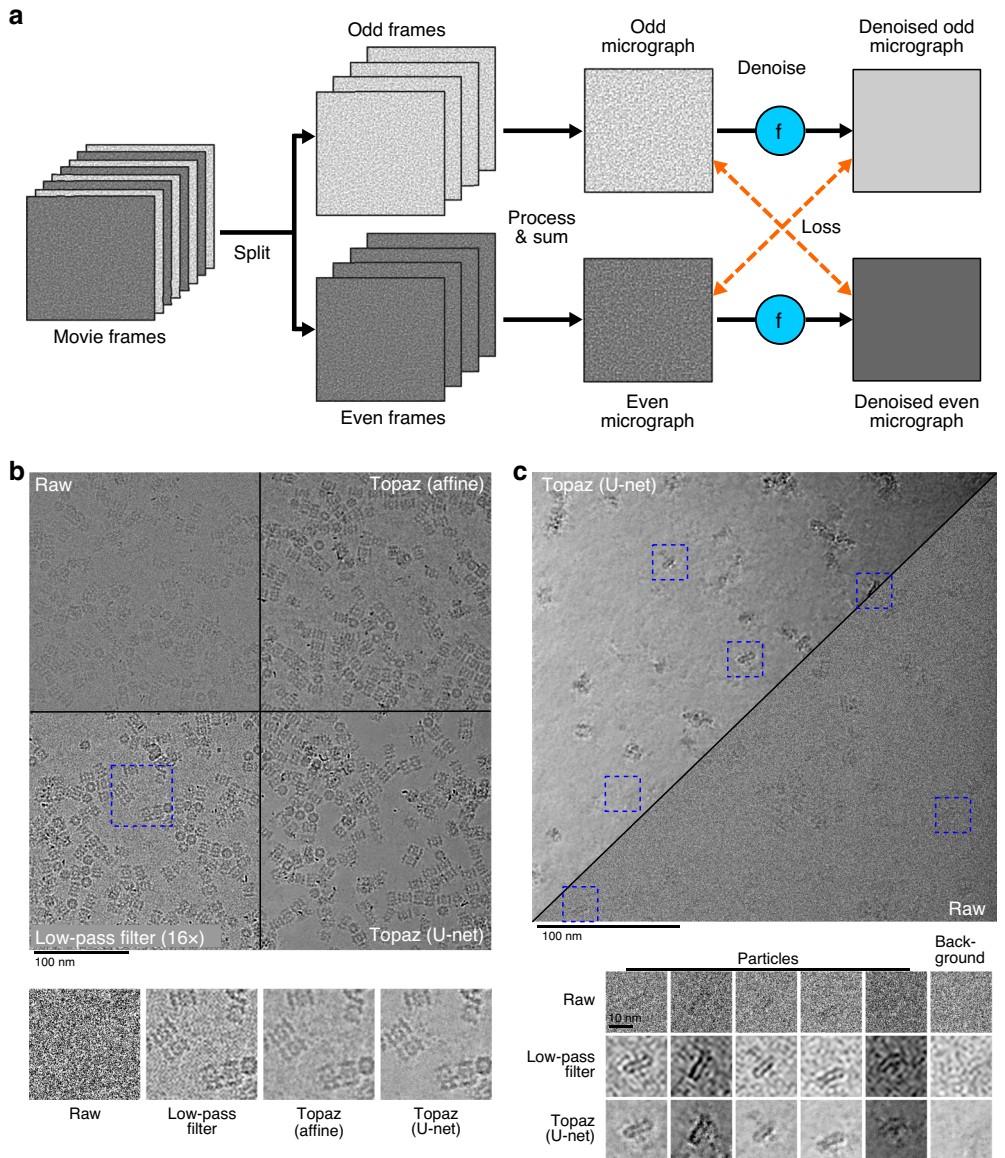

**Fig. 1 Illustration of the training framework and comparison of denoising methods on two example micrographs. a** The Noise2Noise method requires paired noisy observations of the same underlying signal. We generate these pairs from movie frames collected in the normal cryoEM process, because each movie frame is an independent sample of the same signal. These are first split into even/odd movie frames. Then, each is processed and summed independently following standard micrograph processing protocols. The resulting even and odd micrographs are denoised with the denoising model (denoted here as *f*). Finally, to calculate the loss, the odd denoised micrograph is compared with the raw even micrograph and vice versa. **b** Micrograph from EMPIAR-10025 split into four quadrants showing the raw micrographs, low-pass filtered micrograph by a binning factor of 16, and results of denoising with our affine and U-net models. Particles become clearly visible in the low-pass filtered and denoised micrographs, but the U-net denoising shows strong additional smoothing of background noise. A detail view of the micrograph is highlighted in blue and helps to illustrate the improved background smoothing provided by our U-net denoising model. **c** Micrograph from EMPIAR-10261 split into the U-net denoised and raw micrographs along the diagonal. Detailed views of five particles and one background patch are boxed in blue. The Topaz U-net reveals particles and reduces background noise.

model, and micrographs denoised with conventional methods. We chose to measure SNR using real cryoEM micrographs because the denoising models were trained on real micrographs generated under real-world conditions that no software accurately simulates. Due to the lack of ground truth in cryoEM, SNR calculations are estimates (Methods). We manually annotated paired signal and background regions on micrographs from 10 different datasets (Supplementary Fig. 8). We then calculated the average SNR (in dB) for each method using these regions[23]. We present a comparison of four different denoising model architectures (affine, FCNN, U-net (small), and U-net) trained with L1 and L2 losses on either the small or large datasets

(Supplementary Table 1). Note that the L2 affine filter is also the Wiener filter solution. We find only minor differences between L1 and L2 models, with L1 loss being slightly favored overall. Furthermore, we find that the training dataset is important. Intriguingly, the affine, FCNN, and U-net (small) models all perform better than the full U-net model when trained on the small dataset and perform better than the same models trained on the large dataset. The best performing model overall, however, is the full U-net model trained on the large dataset. This model also outperforms conventional low-pass filtering denoising on all datasets except for one, where they perform equivalently (EMPIAR-10005[24]).

A summary comparison is presented in Table 1, where we report SNR results on each dataset for the best overall performing low-pass filter (16x binning) with the L2 U-net trained on the large dataset and the L1 affine model trained on the small dataset. Our pre-trained U-net model improves SNR by >2 dB on average over low-pass filtering and improves SNR by roughly 20 dB (100 fold) over the raw micrographs. The model generalizes well across different imaging conditions, improving SNR on micrographs collected on K2, Falcon II, and Falcon III cameras as well as micrographs collected in super-resolution and counting modes.

To explore the broadness of our general U-net denoising model, we applied the model to several samples across several non-DDD cameras from two screening microscopes and analyzed them visually (Supplementary Fig. 9). The pixelsizes for these datasets are about twice that of the training data and camera hardware binning by two has also been applied. Despite the differing noise characteristics of these cameras relative to the DDD cameras used for training the U-net denoising model, our general denoising model performs well. We see improvements similar to those noted above. Background is reasonably smoothed while the contrast of protein densities is greatly increased in the proteasome and two apoferritin micrographs. The glutamate dehydrogenase micrograph shows slight artifacts around some proteins, but contrast is substantially improved and denoising allows for clear identification of particle aggregates. These improvements demonstrate that our pre-trained denoising model even generalizes well to micrographs collected on screening microscopes and may enable increased cryoEM screening efficiency.

**Denoising with the general model enables more complete picking of difficult particle projections**. We denoised micrographs of particles with particularly difficult-to-identify projections, clustered protocadherin (EMPIAR-10234[25]), to test whether denoising enables these views and others to be picked more completely than without denoising. Figure 2 shows a representative micrograph before and after denoising. Before denoising, many particle top-views were indistinguishable by eye from noise (Fig. 2a, left inset). After denoising, top-views in particular became readily identifiable (Fig. 2a, right inset and circled in green).

We manually picked 1,023 particles while attempting to balance the percentage of side, oblique, and top-views of the particle in our picks. Using these picks, we trained a Topaz[16] picking model as described in the "Methods". The resulting model was used to infer a total of 23,695 particles after CryoSPARC[18] 2D classification, 3D heterogeneous refinement to identify two conformations, and 3D homogeneous refinement using "gold standard" FSC refinement on each conformation (Fig. 2b, right). A closed conformation consisting of 13,392 particles confirmed the previous structure obtained using sub-tomogram alignment (EMD-9197)[25]. A putative partially open conformation consisting of 8134 particles was obtained (Fig. 2b, yellow map), which exhibits a dislocation on one end of the dimer and an increased twist of the whole structure relative to the closed conformation. We confirm that these conformations are not random reconstruction anomalies by repeating the reconstruction process six times independently, all of which produce the same two conformations (Supplementary Fig. 10). In comparison, using only the raw micrographs for initial manual picking, the data owner picked 1540 particles to train a Topaz model as described in Brasch et al.[25] that inferred 10,010 particles in the closed conformation after CryoSPARC 2D classification and 3D homogeneous refinement using "gold standard" FSC (Fig. 2, left). Using Topaz-Denoise to help identify particles manually enabled

**Table 1 Comparison of micrograph denoising methods based on estimated SNR (in dB, larger is better).**

| Method | EMPIAR-10261[48] | EMPIAR-10005[24] | EMPIAR-10025[49] | Protocadherin (K2) | 18sep08d (K2) | 19jan04d (K2) | 19may10e (K2) | 18aug17l (Falcon III) | 18sep06d (Falcon III) | 18sep19l (Falcon III) | Overall |
|---|---|---|---|---|---|---|---|---|---|---|---|
| Affine (Topaz) | 5.49 | 1.29 | 0.72 | 4.83 | 4.51 | 8.87 | 12.02 | 10.65 | 6.90 | 9.15 | 6.44 |
| U-net (Topaz) | 7.17 | 1.72 | 1.07 | 5.94 | 6.06 | 8.43 | 13.07 | 15.17 | 7.37 | 13.24 | 7.92 |
| Low-pass | 5.19 | −0.12 | −0.40 | 4.22 | 3.53 | 6.87 | 9.99 | 9.04 | 6.95 | 8.71 | 5.40 |
| Raw | −17.14 | −20.13 | −24.15 | −14.47 | −15.40 | −11.73 | −5.44 | −6.33 | −3.64 | −5.63 | −12.41 |

SNR was estimated from 20 paired signal and background regions selected for each dataset. In each column, the best performing model is bolded. We report denoising results on aligned micrographs for the NYSBC K2 and Falcon III datasets. All datasets were collected in electron counting modes, except for 18sep06d, which was collected using Falcon III integrating mode. Our U-net denoising model performs best overall and is best on all except for the 19jan04d dataset where our affine denoising model slightly outperforms it. We report low-pass filtering at a binning factor of 16 on all datasets, which we found to give better SNR overall compared to Gaussian low-pass filtering.

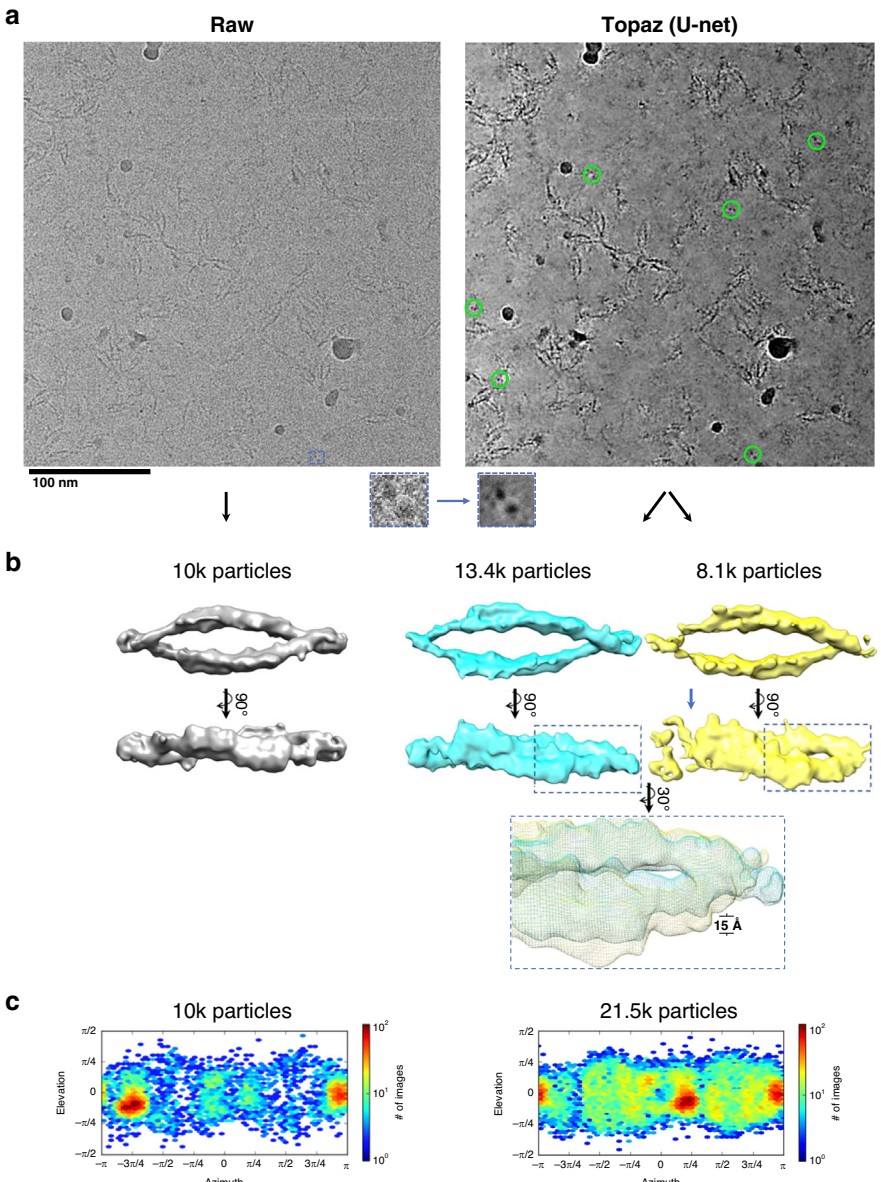

**Fig. 2 Denoising with the general model in Topaz improves interpretability and picking of difficult particle projections. a** A raw micrograph (left) and Topaz-Denoised micrograph (right) of the clustered protocadherin dataset (EMPIAR-10234) with a top-view boxed out (insets). Denoising allows for top-views to be clearly identified (green circles, right) and subsequently used to increase the confidence and completion of particle picking. **b** Topaz picking training on raw micrographs using 1540 manually picked particles from the raw micrographs resulted in the reconstruction on the left. Topaz picking training on the raw micrographs using 1023 manually picked particles from the denoised micrographs resulted in the reconstruction on the right. Manually picking on denoised micrographs resulted in 115% more particles in the 3D reconstruction, which allowed for classification into a closed (blue) and putative partially open (yellow; blue arrow showing disjoint) conformation. The inset shows a zoom-in of the ~15 Å conformational change of the twist. **c** 3D reconstruction particle distributions for (left) Topaz picking training on raw micrographs using 1540 manually picked particles from the raw micrographs, and (right) Topaz picking training on the raw micrographs using 1,023 manually picked particles from the denoised micrographs. All particles from the two classes in (**b**, right) are shown (**c**, right). 3DFSC plots for the three maps shown here are in Supplementary Fig. 14.

us to resolve a putative novel conformation of clustered protocadherin from single particles, resulted in 2.15x more real particles picked, and substantially increased the percentage of top- and oblique-views (Fig. 2c, Supplementary Fig. 11). We substantially improve over the previous best resolution single particle cryoET structure of this protein complex (12 Å vs. 35 Å), yet near atomic resolution single particle structures remain a distant goal. ResLog analysis suggests that millions of particles are required to reach near atomic resolution[26] (Supplementary Fig. 11d).

Interestingly, CryoSPARC ab initio reconstruction using a minimal set of denoised particles is less reliable than using the same set of raw particles (Supplementary Fig. 12). Four or five of the six ab initio reconstructions using the raw particles resulted in the correct overall structure, while only one of the six ab initio reconstructions using the denoised particles resulted in the correct overall structure.

**Denoising with the general model enables shorter exposure imaging.** We simulated short exposure times at the microscope

by truncating frames of several datasets used during frame alignment and summed to the first 10%, 25%, 50%, and 75% of the frames. These datasets were collected with a total dose of between 40 and 69 e-/Å$^2$. We denoised each short exposure with our general U-net model and compared both visually and quantitatively to low-pass filtering and to the raw micrographs without denoising.

Figure 3 shows denoised and low-pass filtered example micrographs of each subset along with the raw micrographs. Visual analysis and our SNR analysis suggests that between 10% and 25% of the exposure time is comparable to the full, raw micrographs (Fig. 3 and Supplementary Fig. 13 for FFT,

Supplementary Figs. 14–17). This corresponds to between 4.0 and 16.7 e-/Å$^2$. 3D reconstructions of frame titrations of identical apoferritin particles from 19jan04d suggests that a total dose of ~16.7 e-/Å$^2$ is required for accurate CTF estimation (Supplementary Fig. 18). Remarkably, reconstructions from these low-dose particle stacks reach resolutions surpassing that of the full dose particle stacks. This suggests that Topaz-Denoise can enable low-dose collection, particle identification, and thus high-resolution reconstruction in practice. Furthermore, roughly double the electron dose is required for low-pass filtering to match the SNR of our neural denoised micrographs. This could allow a factor of two or more savings in exposure time. Such a

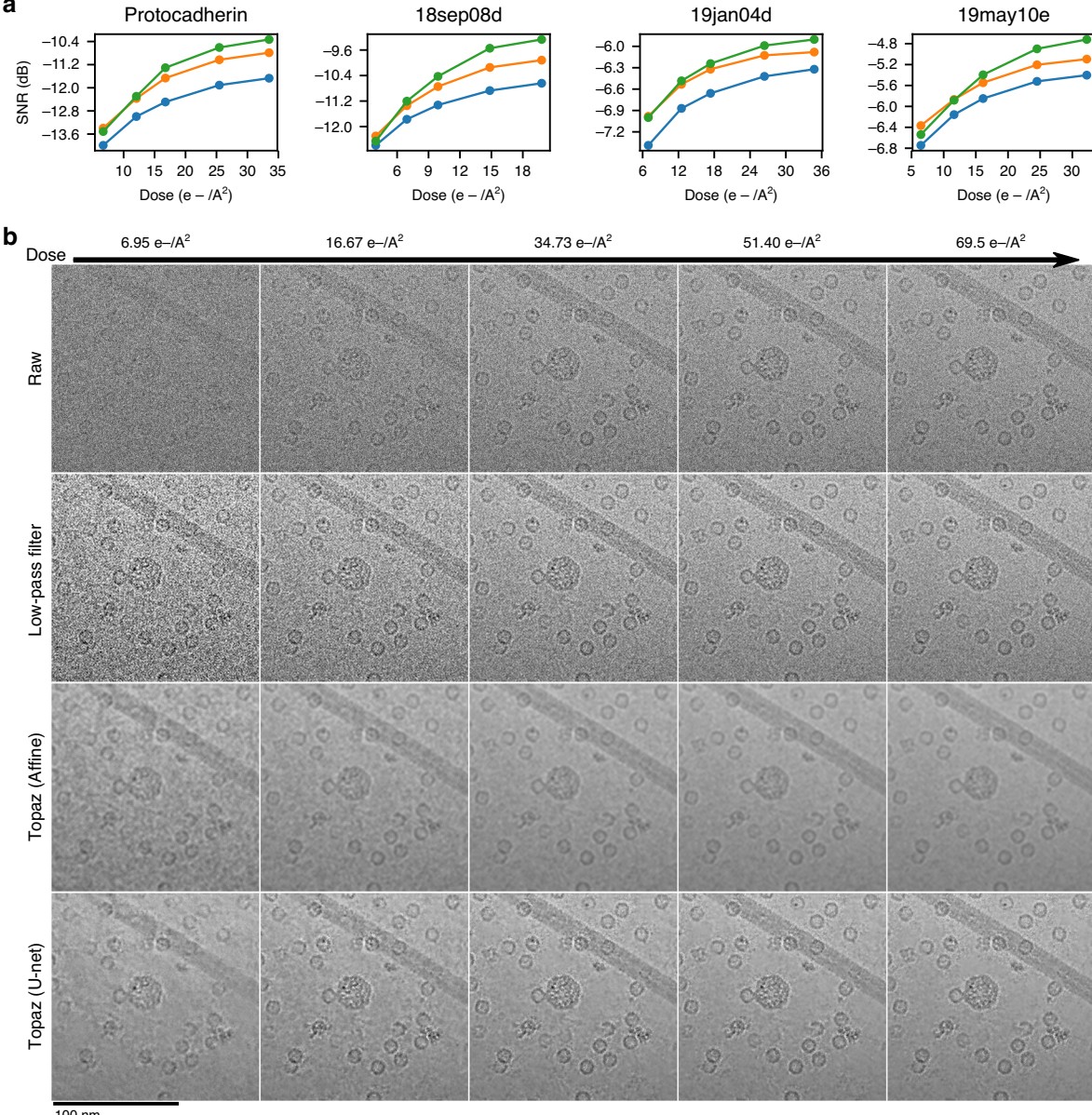

**Fig. 3 Denoising with the general model in Topaz enhances SNR of short exposure micrographs. a** SNR (dB) calculated using the split-frames method (see Methods) as a function of electron dose in low-pass filtered micrographs by a binning factor of 16 (blue), affine denoised micrographs (orange), and U-net denoised micrographs (green) in the four NYSBC K2 datasets. Our U-net denoising model enhances the SNR of micrographs across almost all dosages in all four datasets. U-net denoising enhances SNR by a factor of 1.5× or more over low-pass filtering at 20 e-/Å$^2$. **b** Example section of a micrograph from the 19jan04d dataset of apoferritin, β-galactosidase, a VLP, and TMV (full micrograph in Supplementary Figs. 3 and 4) showing the raw micrograph, low-pass filtered micrograph, affine denoised micrograph, and U-net denoised micrograph over increasing dose. Particles are clearly visible at the lowest dose in the denoised micrograph and background noise is substantially reduced by Topaz denoising.

**Table 2 Comparison between normal dose exposures versus optimized dose exposures using typical high-resolution microscope systems and parameters.**

|  | Exposures/h with 1 exposure/hole (K2) | Exposures/h with 4 exposures/hole (K2) | Exposures/h with 1 exposure/hole (K3) | Exposures/h with 4 exposures/hole (K3) |
|---|---|---|---|---|
| Normal dose (~66 e-/Å²) | 108 | 121 | 195 | 237 |
| Optimized dose (~17 e-/Å²) | 178 | 190 | 242 | 273 |
| % increase in collection efficiency | 65% | 57% | 25% | 15% |

Each entry reports the number of exposures collected using the normal versus denoised optimized doses using either a K2 or K3 camera on a Titan Krios and collecting only one exposure per hole versus four exposures per hole. The optimized dose yields more than a 50% increase in throughput on the K2 camera and still yields a modest improvement in throughput on the newer K3 camera.

significant reduction in exposure time substantially increases the efficiency of cryoEM collection sessions, allowing for microscopes to operate at higher throughput.

To account for real-world collection overhead, we tested the optimal exposure dose for 19jan04d (~17 e-/Å²) compared to a normal exposure dose (~66 e-/Å²) on both a Titan Krios + Gatan K2 system and a Titan Krios + Gatan K3 system for a typical stage shift collection (1 exposure per hole) and a typical image shift collection (4 exposures per hole). Table 2 shows the results. On the K2 system with stage shift collection, using the optimal exposure dose is about 65% more efficient than using the normal exposure dose (178 exposures per hour compared vs. 108). With image shift collection, using the optimal exposure dose is about 57% more efficient than using the normal exposure dose (190 exposures per hour compared vs. 121). On the K3 system with stage shift collection, using the optimal exposure dose is ~25% more efficient than using the normal exposure dose (242 exposures per hour compared vs. 195). With image shift collection, using the optimal exposure dose is ~15% more efficient than using the normal exposure dose (273 exposures per hour compared vs. 237). These results show that using the Topaz-Denoise general model to optimize exposure dose can allow for on the order of 1000 more exposures per day to be collected on K2 and K3 systems.

**Generalized 3D cryoET tomogram denoising markedly improves contrast, SNR, and interpretability.** We converted the 2D Noise2Noise framework used in the previous sections to 3D for the purpose of creating a pre-trained general denoising model for cryo-electron tomograms (Methods). To train a general denoising model, we split 32 aligned cryoET tilt-series from FEI Titan Krios + Gatan K2 BioQuantum systems of cellular and reconstituted biological environments into even/odd frame tilt-series, binned each tilt-series by 2, reconstructed each tilt-series, and trained the neural network for over one month ("Methods"). The average pixelsize of the trained model, called Unet-3d-10a in the Topaz-Denoise package, is 10 Å. To further increase the broadness of 3D denoising in Topaz-Denoise, we trained a second general 3D denoising model called Unet-3d-20a using the same data as the Unet-3d-10a model, except with all training tomograms binned by another factor of 2 in Fourier space (ie. 20 Å pixelsize tomograms). Both general 3D denoising models are included in Topaz-Denoise.

To evaluate the resulting general 3D denoising model, we applied the model to one tomogram from each of the datasets used in the training and compared the results to models trained specifically on each test tomogram ("self-trained"), in addition to low-pass filtered tomograms (Supplementary Table 2). Comparisons were made both by SNR calculations using even/odd tomograms (Supplementary Table 2, "Methods"), and visually.

Our pre-trained 3D U-net model (Unet-3d-10a) improves SNR by >3 dB over raw tomograms and improves SNR by about 1 dB on average over the best low-passed tomograms. Self-trained models showed only a marginal improvement in SNR over Unet-3d-10a. Figure 4a and Supplementary Movie 1 show a visual comparison of one of the yeast tomograms used for training and testing. The Unet-3d-10a and self-trained models show a marked improvement in contrast and detail of ribosomes, RNA, ER proteins, mitochondrial transmembrane proteins, and aggregates over the raw and low-passed tomograms, while flattening background similar to the 2D U-net model for micrographs.

We next applied the Unet-3d-10a model to a sample unlike those it was trained on in several respects: an 80 S ribosome single particle unbinned tomogram with a pixelsize of 2.17 Å and defocus of 4 microns, over four times less than the average pixelsize and half the average defocus of the tomograms used for training. A visual comparison of the applied model along with binned and Gaussian low-pass filtered tomograms is shown in Fig. 4b and Supplementary Movie 2. As with the previous 2D and 3D Topaz-Denoise general model results, the Unet-3d-10a model properly flattens background while increasing contrast of proteins relative to binning and low-pass filters. The increased contrast without tomogram resampling allows for visual delineation of objects of interest while retaining their higher-resolution information, and does not require ad-hoc parameter adjustment or training required by filtering methods more complicated than low-pass filtering. Furthermore, we show that applying a Gaussian filter after denoising further increases contrast, but at the expense of higher-resolution information (Fig. 4b and Supplementary Movie 2, last tomogram). This may be useful if researchers wish to further increase contrast and do not require all frequencies to be visualized.

## Discussion

CryoEM has long been hampered by the ability for researchers to confidently identify protein particles in all represented orientations from behind sheets of noise. Several bottlenecks in the general cryoEM workflow may preclude protein structure determination due to low SNR, such as differentiating protein from noise during picking, picking homogeneous subsets of particles, picking sufficient numbers of particles in all represented orientations, and obtaining a sufficient number of particles for 3D refinement. The initial stages of de novo protein structure determination are particularly affected by these issues.

To ameliorate these potentially critical issues, we present Topaz-Denoise, a Noise2Noise-based convolutional neural network for learning and removing noise from cryoEM images and cryoET tomograms. By employing a network trained on dozens of datasets to account for varying sample, microscope, and collection parameters, we achieve robust general denoising for

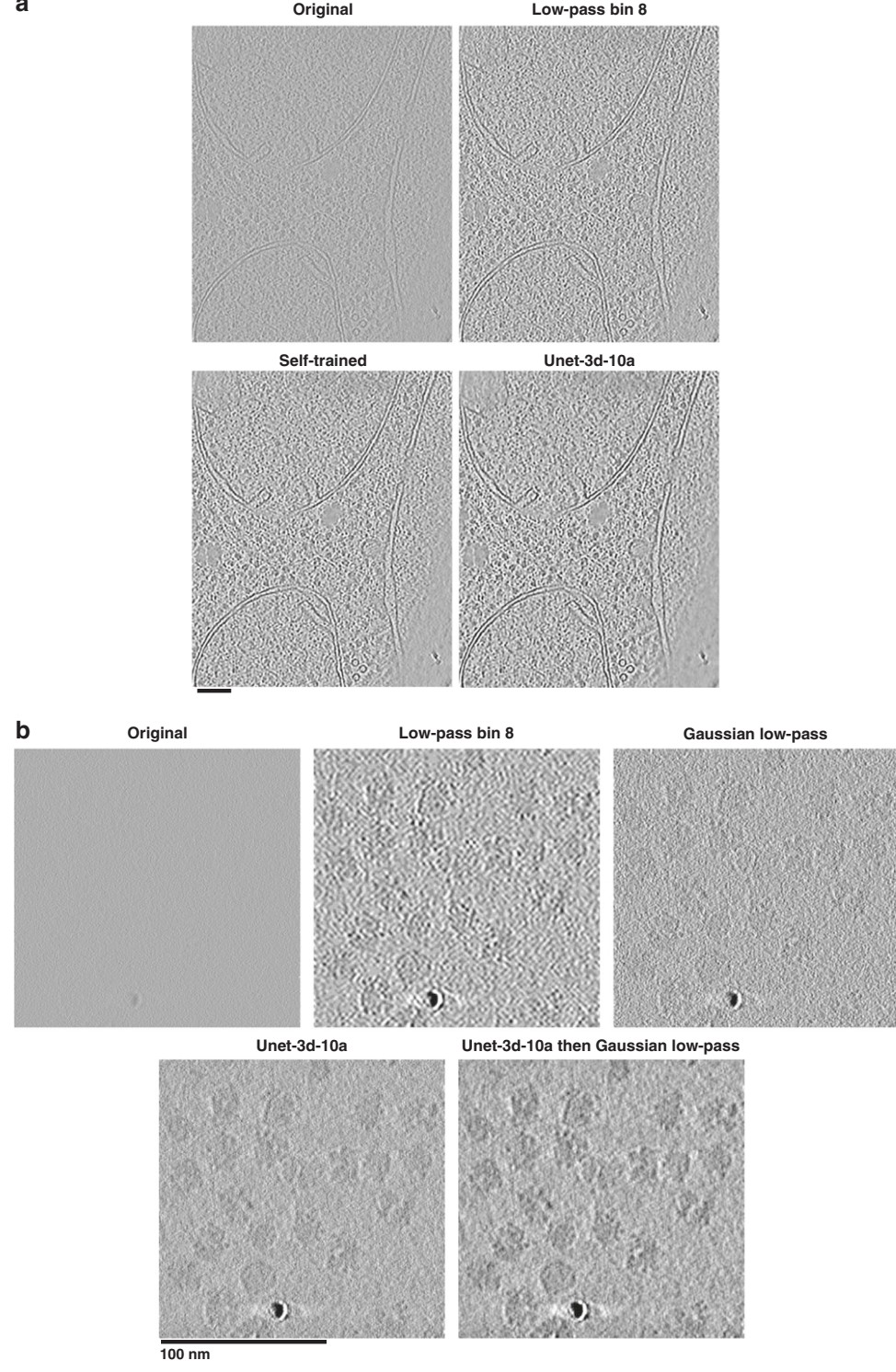

**Fig. 4 Denoising with the 3D general model in Topaz improves cellular and single particle tomogram interpretability. a** Saccharomyces uvarum lamellae cryoET slice-through (collected at 6 Å pixelsize and 18 microns defocus, then binned by 2). The general denoising model (Unet-3d-10a) is comparable visually and by SNR to the model trained on the tomogram's even/odd halves (Self-trained). Both denoising models show an improvement in protein and membrane contrast over binning by 8 while confidently retaining features of interest, such as proteins between membrane bilayers. Both denoising models also properly smooth areas with minimal protein/membrane density compared to the binning by 8. See Supplementary Movie 1 for the tomogram slice-throughs. **b** 80S ribosomes as single particles (EMPIAR-10045; collected at 2.17 Å pixelsize and 4 microns defocus). The general denoising model (Unet-3d-10a) is markedly improved over binning by 8 and the 1/8 Nyquist Gaussian low-pass, both with smoothing background appropriately while increasing contrast and with retaining features of interest at high fidelity, such as the RNA binding pocket in all orientations. The same 1/8 Nyquist Gaussian low-pass applied to the denoised tomogram further improves contrast by suppressing high-frequencies that the user may deem unimportant. See Supplementary Movie 2 for the tomogram slice-throughs.

cryoEM. We show empirically that our U-net denoising models result in higher SNR relative to affine models and low-pass filters. Topaz-Denoise enables visual identification of low SNR particle views, as exemplified by the clustered protocadherin dataset where denoising allows for more representative and complete 3D reconstructions, significantly more particles picked, and a putative new conformation. This putative partially "open" conformation suggests the possibility that assembly of protocadherin *cis*-dimers are preformed on membranes allowing rapid assembly of lattices and triggering of an avoidance signal when two cell surfaces with identical protocadherin complements come into contact. We note that these proteins are known to form flexible complexes in situ[25], but multiple confirmations were not previously identifiable in single particle cryoEM due to the difficulty in analysing these micrographs. Increased confidence in particle identification using Topaz-Denoise enables novel structures to be obtained from cryoEM data due to substantially increased particle picking completeness. Moreover, due to the considerable increase in SNR of denoised single particle micrographs, exposure time may be reduced without sacrificing the ability to pick particles reliably or perform downstream processing, thus enabling an increase in collection efficiency. Finally, implementing the same Noise2Noise-based network in 3D enables denoising of cryo-electron tomograms in minutes. As shown in both cellular tomograms and single particle tomograms, the 3D general denoising model in Topaz-Denoise properly smooths areas without signal and increases contrast of areas with signal without reducing the visual resolvability of features. This results in substantially higher SNR features both visually and quantitatively. Together, the U-net model for cryoEM and the Unet-3d models for cryoET in Topaz-Denoise offer superior denoising capabilities that are broadly applicable across instruments and sample types.

Conceptually, the Noise2Noise method as applied to micrographs in Topaz-Denoise trains a neural network to transform each denoised half-micrograph into the paired raw half-micrograph, and performs this training over thousands of half-micrograph pairs. We note that this is effectively learning an SNR maximizing transformation. This follows from the relationship between the SNR of a micrograph and the correlation coefficient between paired independent measurements, which has been known and used in cryoEM since at least 1975[1,27]. This relationship, $SNR = CCC/(1-CCC)$, where CCC is the correlation coefficient, has direct connection with the Noise2Noise objective in which we seek to find a transformation of the micrograph such that the error between transformed micrographs and the paired independent micrograph is minimized. In particular, the choice of L2 loss can be motivated through direct connection with the correlation. When both the denoised micrograph and raw micrograph are normalized to have mean zero and standard deviation one, the mean squared error (MSE) and correlation coefficient (CCC) are related by $MSE = 2 - 2 \times CCC$. This suggests a direct link between the MSE objective and SNR under the framework of Frank and Al-Ali[27].

Practically in our experience, because the general models were trained on large datasets from popular DDDs (Gatan K2, FEI Falcon II, and FEI Falcon III), these models provide the best visual results on comparable DDDs. For micrographs from microscopes and detectors used in the training dataset, we find that denoising typical featureful objects, such as proteins, continuous carbon, carbon/gold edges, and crystalline ice, increases their visual contrast, while denoising amorphous objects such as vitrified water results in visual flattening (Fig. 1b, c and Supplementary Fig. 13 for FFTs). Micrographs from non-DDD cameras still fare well compared to DDDs in our experience (Supplementary Fig. 9) despite differing physical characteristics of the microscopes and detectors. This suggests that the general U-net

model in Topaz-Denoise is robust to micrographs collected on equipment outside of the training dataset. Since non-DDD cameras are often used on screening microscopes, denoising these micrographs may increase screening throughput by allowing for more rapid analysis of micrographs, thereby increasing the efficiency of grid preparation steps. These results highlight three of the main advantages of our general denoising model: (1) users do not have to spend additional time training specific denoising models using their data, (2) for cameras that do not record frames, such as most screening microscope systems, acquiring data for training is not practical, thus a general denoising model is greatly prefered, and 3) the general model enables real-time denoising during data collection because denoising takes only seconds per micrograph.

The 3D cryoET denoising model included in Topaz-Denoise, and the framework which allows users to train their own models, may allow for improved data analysis not only in the cryoET workflow, but also the cryoEM workflow. In cryoET, researchers are often exploring densely-packed, unique 3D structures that are not repetitive enough to allow for sub-volume alignment to increase the SNR. The 3D denoising model shown here and included in the software increases the SNR of tomograms, which as a consequence may make manual and automated tomogram segmentation[28] easier and more reliable. In single particle cryoEM, we anticipate that the 3D denoising model and models trained on half-maps may be used to denoise maps during iterative alignment, as has previously been shown to be useful after alignment[29]. In our experience, training models on half-maps performs a form of local b-factor correction on the full map, which may allow for more reliable and accurate iterative mask generation during single particle alignment.

Models generated using Topaz-Denoise, including the provided general models, may be susceptible to the hallucination problem in neural networks[30]. Details in denoised micrographs or tomograms may exhibit imprints of encodings from the datasets used for training. Practically, this means that denoised particles should not be relied on for reconstruction as demonstrated in Supplementary Fig. 12. We suspect that the issue here is two-fold: (1) cryoEM/ET refinement and reconstruction software assume noise distributions typical of raw data, not denoised data, and (2) denoised particles may present hallucinated information from the denoising model that is not detectable by visual inspection. For these reasons, we recommend that Topaz-Denoise models be used to assist with visualization and object identification, then the objects of interest be extracted and processed from raw micrographs/tomograms. Misuse of Topaz-Denoise and other opaque augmentations of raw data may result in subtle and difficult-to-detect forms of hallucinated signal[31,32].

As cryoEM and cryoET continue to expand into adjacent fields, researchers new to micrograph and tomogram data analysis will benefit from improved methods for visualization and interpretation of these low SNR data. Topaz-Denoise provides a bridge to these researchers, in addition to assisting those experienced in cryoEM and cryoET. We expect Topaz-Denoise to become a standard component of the micrograph analysis pipeline due to its performance, modularity, and integration into CryoSPARC, Relion, Appion, and Scipion.

## Methods

**Training dataset preparation for 2D denoising models**. To train the denoising models, we collected a large dataset of micrograph frames from public repositories[33] and internal datasets at the New York Structural Biology Center (NYSBC), as described in Supplementary Table 3. These micrograph frames were collected under a large variety of imaging conditions and contain data collected on FEI Krios, FEI Talos Arctica, and JEOL CRYOARM300 microscopes with Gatan K2, FEI Falcon II, and FEI Falcon III DDD cameras at both super-resolution (K2) and counting modes and at many defocus levels. Including several microscopes,

cameras, and datasets allows for robust denoising parameters to be modelled across common microscope setups.

We form two general aggregated datasets, one we call "Large" and one called "Small". The "Large" dataset contains micrographs from all individual datasets. To roughly balance the contribution of the individual datasets in these aggregate datasets, we randomly select up to 200 micrographs from each individual dataset for inclusion rather than all micrographs. The Small dataset contains micrographs from individual datasets selected by eye based on the denoising performance of individually-trained U-net denoising models.

The Noise2Noise framework requires paired noisy observations of the same underlying signal. We generate these pairs by splitting the micrograph frames into even/odd frames which represent independent observations. These even/odd micrograph frames are then summed directly to form the paired observations. Because micrographs are typically motion corrected before summing and this motion correction procedure can change the noise distribution of the micrographs, we also form aligned, summed micrograph pairs by aligning the even/odd micrograph frames with MotionCor2[34] using 5 by 5 patches and a b-factor of 100. This resulted in 1929 paired micrographs for the Small dataset and 3439 paired micrographs for the Large dataset.

**Model architectures**. We adopt a U-Net model architecture[35] similar to that used by Lehtinen et al.[9] except that the input and output feature maps are one-dimensional ($n = 1$ to match monochrome micrographs) and we replace the first two width 3 convolutional layers of Lehtinen et al. with a single width 11 convolutional layer (Supplementary Fig. 19) similar to other convolutional neural networks used in cryoEM[16]. This model contains five max pooling downsampling blocks and five nearest-neighbor upsampling blocks with skip connections between down- and up-sampling blocks at each spatial resolution. We refer to this as the U-net model. For comparison, we also consider a smaller U-net model with only 3 downsampling and upsampling blocks which we refer to as the U-net (small) model. We also compare with a fully convolutional neural network consisting of three convolutional layers of width $11 \times 11$ with 64 filters each and leaky rectified linear unit activations, termed FCNN, and an affine model with a single convolutional filter of width $31 \times 31$.

**Loss functions and the Noise2Noise framework**. The Noise2Noise framework takes advantage of the observation that we can learn models that recover statistics of the noise distribution given paired noisy observations of the same underlying signal. Given a ground truth signal, $y$, we observe images of this signal that have been corrupted by some probabilistic noise process, $x \sim Noise(y)$. Given paired noisy observations for matched signals, $x_a \sim Noise(y)$ and $x_b \sim Noise(y)$, we can learn a function that recovers statistics of this distribution. This is accomplished by finding parameters of the denoising function, $f$ with parameters $\theta$, such that the error between the denoised sample $f(x_a)$ and raw $x_b$ are minimized. The form of this error function determines what statistics of the noise distribution we learn to recover. Given a dataset, $X$, containing many such image pairs, minimizing the L2 error over paired samples,

$$\arg\min_\theta E_{x_s,x_b \sim X}[||f(x_a) - x_b||_2^2], \qquad (1)$$

finds $f$ with mean-seeking behavior. Minimizing the L1 error over paired samples,

$$\arg\min_\theta E_{x_s,x_b \sim X}[||f(x_a) - x_b||_1], \qquad (2)$$

finds $f$ with median-seeking behavior. Finally, minimizing the L0 error over paired samples,

$$\arg\min_\theta E_{x_s,x_b \sim X}[||f(x_a) - x_b||_0], \qquad (3)$$

finds $f$ with mode-seeking behavior. This last objective is not differentiable and requires a smoothing term to minimize with standard gradient descent. We refer the reader to Lehtinen et al.[11] for details on this training objective. In practice, the errors of denoising both $x_a$ and $x_b$ are used for training.

**Training details**. For neural networks, weights are initialized using the default initialization in PyTorch[36]. For affine models, weights are initialized to zero. All models are trained using the Adagrad variant of stochastic gradient descent[24] with a learning rate of 0.001 for 100 epochs. We train on 800 by 800 patches randomly sampled from each micrograph using a minibatch size of 4. As data augmentation, these patches are randomly rotated and mirrored. In order to avoid interpolation artifacts, images are only rotated by 90°, 180°, or 270°. Images are first normalized at the whole micrograph level by subtracting the mean pixel intensity and dividing by the standard deviation of the pixel intensities. Models were trained on a single NVIDIA V100 GPU with 32 GB of VRAM. Training took about 15 h per model.

**Inference details**. Given a trained denoising model, we denoise full-size micrographs. When operating on a GPU, RAM constraints may require denoising to be performed in patches. Here, we denoise in patches of 4000 by 4000 pixels. In order to avoid artifacts that can occur at the patch edges when stitched together, we include padding of 500 pixels around each patch when denoising. We note that this padding approach perfectly resolves patch edge effects when the padding is at least

as large as half the receptive field of the model. Whole micrographs are first normalized by subtracting the mean and dividing by the standard deviation of the pixel intensities. The pixel intensities of the denoised micrograph are then restored by multiplying by the standard deviation and adding back the mean. Given the trained denoising model, inference is fast. We are able to denoise 4k by 4k K2 images at a speed of ~3 s/micrograph on a single NVIDIA 1080 Ti.

**Micrograph scaling for figure visualization**. In order to properly scale the pixel intensities between raw and denoised micrographs for visualization, we scale each micrograph to be relative to the intensities of the 16× low-pass filtered micrograph. This is achieved by subtracting the mean and dividing by the standard deviation of the pixel intensities in the low-pass filtered micrograph. This ensures that the relative signal levels are identical between all processed versions of the micrograph. Furthermore, to convert these into 256-bit values for display, we bin the real number pixel intensities into uniformly spaced buckets between −4 and 4.

**Signal-to-noise quantification**. We quantify the SNR of raw micrographs and processed micrographs in two ways: (1) based on paired labeled signal and background regions, and (2) using two independent measurements of the same signal. For the first method, we hand-labeled 20 signal and paired background regions across up to 10 micrographs from each dataset. We sought to label a variety of signal regions and to select paired background regions as close as possible to each signal region. Labeling was performed with reference to low-pass filtered micrographs in order to prevent any possible bias towards our denoising models. Given $N$ signal, background region pairs, $x_s^i, x_b^i$, indexed by $i$, we calculate the mean and variance of each background region, $\mu_b^i$ and $v_b^i$. From this, the signal for each region pair is defined as $s^i = (\mu_s^i - \mu_b^i)^2$ where $\mu_s^i$ is the mean of signal region $i$. We take the mean over the signal region to be the signal in order to reduce noise and because we sought to label small, uniform signal regions. We then calculate the average SNR in dB for the regions,

$$SNR = \frac{10}{N} \sum_{i=1}^{N} \log_{10}(s^i) - \log_{10}(v_b^i), \qquad (4)$$

which is reported for each dataset given raw and denoised micrographs. This is equivalent to ten times the log of the SNR. We report SNR this way for consistency with the denoising literature.

For the second, independent method of calculating SNR, we adopt the approach of Frank and Al-Ali[27] in which the SNR is estimated from two independent measurements of the same signal using the relationship $SNR = p/(1 − p)$ where $p$ is the cross-correlation coefficient between the two measurements. We use this to measure the SNR of denoising by splitting micrographs into even and odd frames, frame aligning each independently, denoising the odd frame micrograph and then calculating the correlation between the denoised odd frame micrograph and the raw even frame micrograph. Based on this, we report the SNR in dB as

$$SNR = 10\log_{10}(p/(1 − p)) \qquad (5)$$

where $p$ is now the cross-correlation between the denoised odd frame micrograph and the raw even frame micrograph. An advantage of this SNR calculation over our estimate above is that it considers all pixels in the micrograph and does not require labeling signal and background regions. The disadvantage is that it requires paired measurements and, therefore, can only be calculated for datasets where we have the raw frames. We also cannot calculate the SNR of the complete micrograph.

SNR quantification for cryo-electron tomograms is performed using the second method, where a ~$1000 \times 1000 \times 150$ pixel sub-volume of each tomogram containing biological objects is used to calculate SNR.

**Short exposure micrograph processing**. To quantify our ability to improve interpretability of low electron dose micrographs, we selected between five and ten random micrographs for the four datasets presented (EMPIAR-10234, 18sep08d, 19jan04d, and 19may10e). Micrographs from each dataset were split into five frame-fractionated subsets using IMOD's newstack program[37] to simulate short exposures: 10%, 25%, 50%, 75%, and 100%. Frames were aligned with Motioncor2 using $5 \times 5$ patches and dose weighting. For each dataset, SNR quantification was performed using the second method described above. For this quantification, micrographs were split into even and odd frames. The odd frames were then dose fractionated into 10%, 17.5%, 25%, 37.5%, and 50% of the full-micrograph total doses (rounded down to the nearest frame) and denoised. These low-dose denoised micrographs were compared with their corresponding full-dose even frame micrograph to calculate the SNR. The full-micrograph total doses for EMPIAR-10234, 18sep08d, 19jan04d, and 19may10e are 67.12 e-/Å², 39.6 e-/Å², 69.46 e-/Å², and 64.44 e-/Å², respectively.

**Short exposure apoferritin processing**. To quantify downstream results from frame titration, 100 random independently frame-aligned fractionated micrographs of 19jan04d were prepared using Motioncor2 without dose weighting. CTF estimation of the resulting 500 frame aligned micrographs was performed using CTFFind4[38] from within Appion[20]. 9,373 particles were picked from the micrographs using the first 10% of frames (the first 1 s of the exposures), an initial model

was created in CryoSPARC, and the particles were refined through homogeneous refinement using "gold standard" FSC. The same particle picks and initial model were then used to extract and process the 25%, 50%, 75%, and 100% subsets through de novo 3D homogeneous refinement using "gold standard" FSC while retaining each independent micrograph CTF estimation. 3DFSC[39] plots were then generated from the results.

**Optimized Krios K2, K3 exposure time test**. To quantify impacts on collection time using exposure doses optimized with Topaz-Denoise, we set up several collection sessions on FEI Titan Krios + Gatan K2 and FEI Titan Krios + Gatan K3 systems running Leginon collection. For each system, we used an apoferritin grid with 1.2/1.3 hole spacing, 300 mesh and collected four datasets to mimic common collection parameters: one with one normal exposure (~66 e-/Å$^2$) and one focus per hole, one with one optimized exposure (~17 e-/Å$^2$) and one focus per hole, one with four normal exposures (~66 e-/Å$^2$) and one focus per hole, and one with four optimized exposures (~17 e-/Å$^2$) and one focus per hole. On the K2 system, normal exposures took 10 s and optimized exposures took 2.5 s. On the K3 system, normal exposures took 2.5 s and optimized exposures took 0.65 s. Before each exposure was collected, the stage was set to settle for 1 s. Collection for each condition lasted 20-30 min to account for average overhead time during collection (focusing, stage movement, camera readout, etc.). Collection times were then extrapolated to exposures per hour and exposures per day. Additional overhead such as LN2 re-filling and targeting optimal sample areas were not taken into account.

**EMPIAR-10234 clustered protocadherin single-particle processing**. We processed the EMPIAR-10234 clustered protocadherin dataset in two separate ways to test whether picking in denoised micrographs was advantageous: First by using the particle picks provided by the data owner, and second by manually picking on the denoised micrographs.

The picking method used by the data owner is described in Brasch et al.[25] Briefly, 1540 particles were manually picked by the data owner from 87 raw micrographs and used to train a Topaz[16] picking model, resulting in 14,569 particles. The following reconstruction workflow was performed in CryoSPARC v2[18] using C1 symmetry in every step and using frame-summed particles for consistency. 2D classification was performed three times to remove obvious non-particle classes, resulting in 13,739 particles. Ab initio reconstruction with two classes was performed, resulting in one good class with 10,010 particles. 3D homogeneous refinement using "gold standard" FSC was performed resulting in the final reconstruction.

The picking method we used is as follows: frame-summed micrographs were denoised with the Topaz-Denoise v0.2.1 L2 model, proprocessed with "topaz preprocess" while binning by a factor of 4, and 1023 particles were manually picked not by the data owner from 215 denoised micrographs. A Topaz[16] picking model was trained using the particle coordinates on raw micrographs, resulting in 59,273 particles. The following reconstruction workflow was performed in CryoSPARC v2 using C1 symmetry in every step and using frame-summed particles for consistency. 2D classification was performed three times to remove obvious non-particle classes, resulting in 44,303 particles. Ab-initio reconstruction with two classes was performed, resulting in one good class with 23,695 particles. Heterogeneous refinement with 2 classes was performed, resulting in two classes with different conformations. 2D classification and heterogeneous refinement with these two classes and a junk class were performed, resulting in 13,392 particles in the closed conformation and 8134 particles in the partially open conformation. 3D homogeneous refinement using "gold standard" FSC was performed on each class, resulting in the final reconstructions. UCSF Chimera[40] was used to render the final reconstructions.

**Clustered protocadherin low particle number single-particle processing**. Denoising and picking were performed as described in the last paragraph of the previous section. Then 1000 random particles were chosen and processed through CryoSPARC v2 ab initio reconstruction six times using the raw particles and six times using the particles denoised by the v0.2.1 L2 model. Comparisons between the full 3D map and each set of six ab initio models were made in UCSF Chimera[40].

**Cryo-electron tomography and 3D denoising**. *Saccharomyces uvarum* (*S. uvarum*) was grown to log phase in YPD media, applied to freshly glow discharged carbon/copper 200 mesh Quantifoil grids, back blotted with the Leica GP, and plunge-frozen into liquid ethane. Grids were clipped and loaded into a Helios NanoLab 650 dual beam. The milling protocol[41] yielded lamellae ranging in thickness between ~100–150 nm. Briefly, yeast cells were plunge frozen after back-blotting using an FEI Vitrobot, a conductive platinum layer was deposited in a Quorum cryo-stage, a protective layer of platinum GIS was deposited, FIB-milling was performed in an FEI Helios NanoLab 650, and a final layer of conductive platinum was deposited onto the lamellae. Tomograms were collected with a FEI Titan Krios equipped with an energy filter and a Gatan K2 BioQuantum operated in counting mode with the following parameters: energy filter 30 eV slit, ~18 μm defocus, 6.08 Å pixelsize, −48.0° to 48.0° tilt with a 2° increment, total dose of ~60 e-/Å$^2$. Tomograms were collected with a bi-directional scheme with an unreleased hybrid track/predict method[42] implemented in Leginon[43]. Raw frames were aligned with MotionCor2[34] and split into even/odd pairs before summation. Whole

tilt-series were aligned with Appion-Protomo[44,45], then split into even/odd frame tilt-series, and reconstructed with Tomo3D[46,47] weighted back-projection, yielding even/odd half tomograms. Thirty two tomograms of cellular and reconstituted environments collected on FEI Titan Krios + Gatan K2 BioQuantum systems with an average pixelsize of 5.2 Å (range: 1.8–6 Å) and defocus of 9 μm (range: 4–18 μm) were used for training the pre-trained general denoising models, which was performed in parallel across seven Nvidia GTX 1080 GPUs. Training the model for binned-by-two tomograms (Unet-3d-10a) took over one month and training the model for binned-by-four tomograms (Unet-3d-20a) took ten days. Denoising a 3GB binned-by-two tomogram takes about five minutes on an Nvidia RTX Titan GPU and an Nvidia RTX 2080 Ti GPU in parallel. Tomography Figures and Supplementary Movies were made with Imod[37].

The tomogram denoising model uses a U-net architecture identical to the 2D U-net presented above, except that the 2D convolutions are replaced with 3D convolutions to operate on the tomogram voxel grids and the first convolutional kernel width is 7 voxels rather than 11. Training and inference methods are otherwise identical except for the patch size, which is set to 96 for tomograms to fit in GPU RAM.

**Reporting summary**. Further information on research design is available in the Nature Research Reporting Summary linked to this article.

## Data availability

The general models used in this paper are included as options in Topaz-Denoise. 110 NYSBC dataset frames used for some of the models have been deposited to EMPIAR-10473. The closed and partially open clustered protocadherin maps have been deposited to EMD-22060 and EMD-22059, respectively. The apoferritin maps for 10%, 25%, 50%, 75%, and 100% of the total dose have been deposited to EMD-22052, EMD-22053, EMD-22056, EMD-22057, and EMD-22058, respectively. Other data are available from the corresponding authors upon reasonable request.

## Code availability

Source code for Topaz-Denoise is publicly available as part of Topaz (v0.2.0 and above for 2D denoising, v0.2.4 and above for 3D denoising) on GitHub at https://github.com/tbepler/topaz. Topaz is installable through Anaconda, Pip, Docker, Singularity, SBGrid, and source. Topaz is licensed under the GNU General Public License v3.0.

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

## Acknowledgements

We wish to thank Simons Electron Microscopy Center (SEMC) OPs for many of the test datasets used in training and the authors of the EMPIAR entries listed in Supplementary Table 3 for additional training datasets. We wish to thank Dr. Julia Brasch for sharing her processing experience with EMPIAR-10234 and providing opinions on the work in this paper. We wish to thank Dr. Sargis Dallakyan for integrating Topaz-Denoise into Appion, the CryoSPARC team for integrating Topaz-Denoise into CryoSPARC, Dr. Rafael Fernandez-Leiro for help with integrating Topaz and Topaz-Denoise into RELION, and Dr. Grigory Sharov and the Scipion team for integrating Topaz-Denoise into Scipion. We wish to thank Dr. Michael Rout for the yeast specimen used for 3D cryoET denoising. We wish to thank Drs. Anchi Cheng, Mykhailo Kopylov, Bridget Carragher, and Clinton Potter for helpful discussions. This work is dedicated to the memory of K.K. T.B. and B.B. were supported by NIH grant R01-GM081871. A.J.N. was supported by a grant from the NIH National Institute of General Medical Sciences (NIGMS) (F32GM128303). The cryoEM work was performed at SEMC and National Resource for Automated Molecular Microscopy located at NYSBC, supported by grants from the Simons Foundation (SF349247), NYSTAR, and the NIH NIGMS (GM103310) with additional support from the Agouron Institute (F00316) and NIH (OD019994).

## Author contributions

T.B., A.J.N., and B.B. conceived of this project. T.B. developed and implemented the models. K.K. prepared and collected the cryo-FIB/SEM-ET data and implemented 3D denoising. T.B., K.K., and A.J.N. processed and analyzed the data and model results. T.B. and A.J.N. processed and analyzed single particle results. T.B., K.K., A.J.N., and B.B. gave input on the models, designed the experiments and wrote the paper.

## Competing interests

The authors declare no competing interests.
