## [Peer Review File · Nature Communications]

Reviewers' comments:

Reviewer #1 (Remarks to the Author):

The manuscript by Bepler et al. entitled "Topaz-Denoise: general deep denoising models for cryoEM" introduces a deep learning-based denoising method specifically for cryo-EM data. Given the other existing studies (like ref 16), I am overall convinced that the deep learning-based method can very likely do a better job than conventional approaches in denoising cryo-EM data. However, I would like to delay my recommendation until the authors address the following questions:

(1) The deep learning model is insufficiently described. The author claimed that they used a U-Net model architecture similar to that used by Lehtinen et al, but the details about the "single width 11 convolutional layer" is missing. What is the size of other two dimensions of each layer? Are they all the same or changing over layers? The author should provide a clearcut network architecture drawing in either main text figure or Supplementary figure.

(2) What is the rationality of have 11 single-width convolutional layers?

(3) What is the advantage of the authors' network compared to the Noise2Noise framework?

(4) Have the authors trained their network by differentiating data from different types of electron detection devices: (1) Direct detector device (DDD), (2) CCD/CMOS camera, and (3) conventional film? Although the films have been largely out of date and barely used nowadays, the CCD/COMS cameras are widely deployed in low to middle ends of cryo-TEMs at least for sample screening. So, it will be really helpful to know if Topaz-Denoise can perform as well for both DDD and CCD.

(5) Following the above question, the authors seem to be not aware of or at least not make it clear that some of the testing data were taken by CCD but not DDD. Thus, some of the interpretation about the results might have ignored the potential performance difference between CCD and DDD. In fact, we knew that DDD and CCD noises arise from very different physical mechanisms. One is from counting error, while the other comes from inelastic scattering and diffused electrons in the fluorescent layer that ensure long-range crosstalk. How did authors manage to denoise CCD data vs. DDD data?

(6) Still following the above two concerns, the authors should explicitly discuss/explain their treatment for noises from different devices, in term of training and general usage.

(7) In Figure 3b, the 3D reconstructions with and without Topaz-denoise do not make a clearcut difference in terms of the model/map quality. I am not sure that a point can be made herein or if the difference justifies the acclaimed "improvement".

(8) There is also a lack of numerical performance characterization of the algorithm with respect to varying SNRs in the data. To what SNR the algorithm will equally well perform? How will the background look like with even lower SNR after denoising? This can be readily done with simulated noisy micrography with pre-set SNRs, so that the blind test can be compared to the ground truth.

(9) One suggestion for authors to consider: besides particle selection and verification, a very helpful use of the denoising tools is high-throughput sample screening, at which step researchers look at the fewer raw micrographs and decide if those are good for scale-up data collection. Lower contrast sometimes can confuse researchers in their judgement about the goodness of the screen sample. With a more reliable denoising tool, the screening could be made more efficiently. The authors could capitalize on this usage for enhancing the significance of the work.

(10) The last but not the least is the originality and advancement of this present study seems to be rather limited and incremental. The authors seem to make the least changes to U-Net that was the work of someone else, but merely applied it to the cryo-EM data denoising. While this is certainly a welcoming addition to the present portfolio of software tools in cryo-EM data analysis, the step it meant to tackle might be too small to appeal for its significance in justifying its publication in a journal like Nature Communication. I am all ears to hear any additional evidence to justify its importance.

In summary, the authors borrowed an existing CNN architecture called U-Net developed by others previously and applied it to denoise cryo-EM data. Although this is an interesting avenue, the

presented study seems to be incomplete, somewhat premature to be published, and lack of numerical performance characterization with differential SNRs to justify its advantage, novelty and significance. However, if the authors can fully address the above concerns, I am glad to take a fresh look at the manuscript.

Reviewer #2 (Remarks to the Author):

The authors present an interesting application of a denoising algorithm to cryoEM micrographs. The tool is useful as shown by the authors in their various examples. However, the algorithm is almost trivial given the state of the knowledge at this moment. We recommend the publication of the software as an application note in a journal closer to its final users.

Reviewer #3 (Remarks to the Author):

This paper introduces and evaluates a method for cryo-EM micrograph denoising based on deep learning and the recent Noise2Noise learning framework. Specifically, the paper identifies the applicability of the Noise2Noise framework based upon the observation that movie frames in cryo-EM can be approximately viewed as noisy samples of the same underlying "clean" signal. This framework is then used with a variety of network architecture choices and some variations of the specific training loss used.

The proposed denoising algorithm is evaluated for the purposes of visualization and manual particle picking. Quantitative comparisons based on a variation of SNR are also provided. Both quantitative and qualitative results show that the method generally performs well, improving SNR in almost all cases.

The potential of the method is shown by manual selection of particle sets from the denoised micrographs which are then used to train a further automatic particle picker. The results on a challenging dataset (clustered protocadherin) show that this can help in the selection of different particle views for some structures, leading to improvements in resolution and map quality.

Overall it is a solid piece of work and likely to be of interest to cryo-EM practitioners.

Issue: Appropriate Uses of Denoising

=====

I think it's important to note that denoising in cryo-EM can be misused in potentially dangerous ways. The use of denoising as a tool for visualization, exploration and annotation is generally safe and even an ideal application. However, I would very strongly suggest that denoising never be used as part of a reconstruction process. Specifically, denoised micrographs should never be used to perform a reconstruction or refinement. This is because of the increased risk of hallucinated structures and misleading results. It is generally possible that learning-based denoising methods (like the one proposed here) can hallucinate information in the denoised images for a variety of reasons and it is generally not possible to detect. In the context of reconstruction this can lead to a very subtle and hard to detect form of the "Einstein from Noise" phenomena.

Further, most reconstruction/refinement methods are developed assuming that inputs exhibit the noise which is typically present in cryoEM micrographs and are very likely to fail in both obvious and subtle ways with denoised particle images as inputs. This was, indeed, noted by the authors when they observed that cryoSPARC struggled to do a reconstruction on the denoised particles as shown in Fig 15.

I would strongly suggest that a discussion of when and where denoising is appropriate or inappropriate be added to the manuscript as the dangers may not be obvious to many practitioners. In addition, discussion of Supp Figure 15 should be focused on emphasizing why denoised images shouldn't be used for reconstruction.

Smaller Issues

=====

- References 6-8 are an odd collection of denoising papers. I would suggest just citing 6 as a review article of classical methods, finding a more recent reference for learning based methods and dropping 7,8.
- Relatedly, denoising has only recently become of strong interest in the ML community. Previously it was primarily of interest in signal processing and computer vision communities so the statement in the manuscript should be correct.
- I would recommend adding a small figure with the specific network architecture used to make it clearer.
- When training do you only use $||f(x_a) - x_b||$ in the loss or do you also use $||f(x_b) - x_a||$? I imagine you use both but it's not clear from the text and it seems to imply that only one is used. Ideally both would be used.
- When augmenting the data, how is rotation done? The interpolation that is used during rotations can sometimes dampen noise. Was there any special care taken in this step or is it a simple bilinear interpolation operation in image space?
- During inference for large images, how are the overlapping regions combined?
- There are some missing details in the SNR calculation. It also seems like a somewhat strange way to compute the SNR and I think it needs further motivation/justification. I believe in the paper, the magnitude of the "signal" in this case is being defined as the squared difference between the mean value in a signal region and the mean value in a background region. This is likely fine for comparing methods but doesn't really give a true sense of the "SNR" of the denoised image in the sense that the metric is commonly used in image processing.
- In Supp Figure 14, both plots should have the same range for the Y axis to make them more comparable. This will also make it clearer that the larger particle set was helpful.

Reviewer #4 (Remarks to the Author):

The authors adapted a generalized algorithm for removing unwanted features from images, Noise2Noise, to denoise cryo-electron micrographs, calling it "Topaz-Denoise". It uses the current technology of movies or dose-fractionation to construct two versions of each micrograph. A convolutional neural network is then trained on a subset of the data. The signal is then taken as the common part and the noise the difference. To calculate the enhancement in contrast, a signal-to-noise ratio is calculated as the difference in averages between pairs of particles and background regions, divided by the variance (of the signal or background?).

It is an interesting application to enhance the contrast, and is of general interest in image processing. My impression is that it will be mostly useful in picking particles in cryoEM, and may change the way we approach the task. However, care should be taken to emphasize that the denoised particles are not suitable for further processing to generate 3D reconstructions.

One general concern is that the SNR is oddly defined (page 10). The typical definition is the ratio of foreground and background variances. Why was a different definition chosen? I would encourage the authors to adopt a definition for the SNR more in line with that commonly used in cryoEM.

Page 2:

"Trained on thousands of micrographs across a variety of imaging conditions, these models provide robust denoising without the need to train on a dataset-by-dataset basis."

Does this mean that the trained coefficients from this study could be used in any other case? This would avoid the need for further training.

Page 4:

"Furthermore, we find that the training dataset is important. Intriguingly, the affine, FCNN, and U-net (small) models all perform better than the full U-net model when trained on the small dataset and perform better than their equivalents trained on the large dataset."

I don't understand what the "equivalents" are. Please clarify.

Page 5:

"CryoSparc ab-initio reconstruction using a minimal set of denoised particles is less reliable than using the same set of raw particles (Supplemental Figure 15). Four or five of the six ab-initio reconstructions using the raw particles resulted in the correct overall structure, while only one of the six ab-initio reconstructions using the denoised particles resulted in the correct overall structure"

This is an interesting observation that should feature more prominently in the discussion. I'm not surprised that the denoised particles fared worse. The denoising inevitably removes particle features together with the noise (as does low-pass filtering). It should be more clearly emphasized that the denoised particles should not be used for alignment and reconstruction. It would also be interesting to understand how the denoising changes the spectral characteristics as shown in Supplemental Figure 9.

Page 7:

The discussion is very short and only covers generalities. This would be the appropriate place to discuss what happens during the denoising procedure and why it enhances the SNR. Also, what happens with other features such as crystalline ice and carbon edges, and how are these handled?

Page 10:

In the equation for the SNR I presume v_i refers to the background variance for region i . Please clarify.

Response to Reviewers (Beppler, et al.)

We would like to thank the reviewers for their analysis and comments, which have significantly improved our manuscript. In addition to addressing the reviewers' concerns below, we would like to point out that Topaz-Denoise is now integrated into CryoSPARC, RELION, and Scipion, in addition to Appion, significantly boosting its reach and usability for researchers.

We provide individual responses to reviewers' comments and questions in the following pages. We would like to first highlight a number of improvements made based on the comments of the reviewers:

1) ***Topaz-Denoise is now more accessible to users due to integration with CryoSPARC, RELION, and Scipion:***

CryoSPARC in particular includes a full tutorial for Topaz use inside of CryoSPARC:

<https://cryosparc.com/docs/tutorials/topaz/>

CryoSPARC and RELION are the two most popular cryoEM processing suites and Scipion is a popular cryoEM collection and processing suite. With Topaz-Denoise's recent integration, more users will incorporate Topaz-Denoise into their standard cryoEM workflows. This increase in accessibility should significantly increase the impact of this work.

2) ***Expanded on the protocadherin reconstructions enabled by Topaz-Denoise.*** We are now able to resolve two structural states of the protocadherin dimer, a closed state and a novel partially open state, as well as improve resolution of the closed state over the original cryoEM and cryoET structures. The never-before-seen open state suggests a mechanism for clustered protocadherin assembly in solution and on membranes, which is their primary functional location *in vivo*. We now highlight this successful use of Topaz-Denoise for novel structure determination in the Results and Discussion. This new result not only impacts clustered protocadherin studies, but this example serves as a model case of using Topaz-Denoise to assist structure determination of all particles hampered by low SNR in cryoEM and demonstrates the impact of this work significantly.

3) ***3D denoising has been expanded into a separate Results section and a pre-trained 3D denoising model is now included in Topaz-Denoise.*** We have added and analyzed a new 3D denoising model trained on several dozen cryoET tomograms of biological environments. We compare results of the pre-trained model to models trained on individual tomograms. This additional Results section and software functionality expand the usefulness of Topaz-Denoise into the realm of cryo-electron tomography, a

field where object identification in crowded environments is cumbersome and inaccurate. With this addition, we see a marked increase in impact of this work.

4) ***Novelty of methodological advance.*** Bringing techniques developed in the field of machine learning, convolutional neural nets and in particular U-Net, to cryoEM is a worthy conceptual advance in and of itself; many top computational biology publications leverage fundamental methods developed in other fields to solve novel problems. This is especially so given that these techniques enable capabilities and results not possible before.

Point-by-point responses (original comments in black; responses in blue).

Reviewer #1:

Remarks to the Author:

The manuscript by Bepler et al. entitled “Topaz-Denoise: general deep denoising models for cryoEM” introduces a deep learning-based denoising method specifically for cryo-EM data. Given the other existing studies (like ref 16), I am overall convinced that the deep learning-based method can very likely do a better job than conventional approaches in denoising cryo-EM data. However, I would like to delay my recommendation until the authors address the following questions:

(1) The deep learning model is insufficiently described. The author claimed that they used a U-Net model architecture similar to that used by Lehitnen et al, but the details about the “single width 11 convolutional layer” is missing. What is the size of other two dimensions of each layer? Are they all the same or changing over layers? The author should provide a clearcut network architecture drawing in either main text figure or Supplementary figure.

We apologize for the lack of detail. This layer is a width 11, height 11 convolutional layer with one filter. We now include an illustration of the architecture to clarify (Supplemental Figure 18).

(2) What is the rationality of have 11 single-width convolutional layers?

As clarified above, this is a single convolutional filter of width 11. We made this alteration as a simplification of the original architecture to favor rapid downsampling over learning a more complex non-linear filter at the highest pixel resolution. We have now explained this choice in the Methods.

(3) What is the advantage of the authors’ network compared to the Noise2Noise framework?

The primary advantage of the pre-trained general denoising model is that the user does not need to re-train the model using their data, resulting in a faster, less-specialized pipeline. We show in the new 3D denoising results (section 4) that the pre-trained 3D denoising model performs comparably to models trained on individual tomograms. The pre-trained model is also necessary to enable real-time denoising during data collection. Moreover, some data sources do not include frames, such as most screening microscopes. Without movie frames, specialized data collection would be required to apply Noise2Noise. Our general

model can be applied to these datasets directly and it performs well in our tests (see Supplemental Figure 9). We have added these points to the Discussion.

(4) Have the authors trained their network by differentiating data from different types of electron detection devices: (1) Direct detector device (DDD), (2) CCD/CMOS camera, and (3) conventional film? Although the films have been largely out of date and barely used nowadays, the CCD/COMS cameras are widely deployed in low to middle ends of cryo-TEMs at least for sample screening. So, it will be really helpful to know if Topaz-Denoise can perform as well for both DDD and CCD.

All of the datasets presented and used in this manuscript were collected on DDD cameras. We have clarified this in the Introduction and we have updated Supplemental Table 1 to explicitly report on the camera used for each dataset used in training.

CCD data was not included as training data because we do not have access to cryoEM CCD data collected with movies; there are no such public datasets and our institutions do not have CCD cameras that collect movies. Without CCD movie data, we are unable to practically obtain the paired images of identical signal required for training. For the same reason, a sufficient amount of film data with multiple exposures per field of view does not exist and would need to be collected, which is also not practically possible.

That said, we agree that improving the interpretability of micrographs could be very helpful for screening which we now include in the Discussion. To this end, we now report denoising results using our pre-trained model on several non-DDD datasets. Remarkably, our denoising model performs quite well on these datasets despite being trained only on DDD data. We have added several examples of the denoising model applied to non-DDD data (collected with scintillator-based cameras) in the Results, Discussion, and Supplemental Figure 9.

(5) Following the above question, the authors seem to be not aware of or at least not make it clear that some of the testing data were taken by CCD but not DDD. Thus, some of the interpretation about the results might have ignored the potential performance difference between CCD and DDD. In fact, we knew that DDD and CCD noises arise from very different physical mechanisms. One is from counting error, while the other comes from inelastic scattering and diffused electrons in the fluorescent layer that ensure long-range crosstalk. How did authors manage to denoise CCD data vs. DDD data?

We have verified that none of the datasets originally used in this work were collected on CCD cameras. All were collected on modern direct detector cameras (K2, Falcon II, and Falcon III) as described in the Methods section. We now clarify this in the Introduction, Discussion, and Methods. We agree that the large difference in noise characteristics of CCD cameras may limit the application of our DDD trained model to CCD images, but we observe that denoising non-DDD micrographs with the pre-trained model still performs well. As mentioned above, we have added several examples to the Results, Discussion, and Supplemental Figure 9. It would be ideal to train dedicated CCD denoising models, but, unfortunately, because CCD cameras do not collect movie frames, training for CCD cameras would require specialized data collection to take paired images of the same view. Here, we avoid specialized data collection by exploiting the multiple frames normally collected by DDD cameras.

(6) Still following the above two concerns, the authors should explicitly discuss/explain their treatment for noises from different devices, in term of training and general usage.

We learn a model that is robust to noise properties of the different DDD cameras by including images collected by these different cameras and on different microscopes in our large aggregate training set. We then show that this general model denoises images collected with these cameras in several new datasets (Table 1) without retraining. We now emphasize this point in the Results.

(7) In Figure 3b, the 3D reconstructions with and without Topaz-Denoise do not make a clearcut difference in terms of the model/map quality. I am not sure that a point can be made herein or if the difference justifies the acclaimed “improvement”.

We thank the reviewer for pointing this out. We have gone back and further processed the data to find that there are actually two conformations of particles present in Figure 3b: one closed and one partially open. Furthermore, by re-classifying the initial particle set, we have found an additional 5.5k particles in the consensus map. This is the first time these structures have been resolved from single particle EM data, and the first time the partially open conformation has been observed by any method. Before, the closed state was only observed by cryoET of the same sample. Remarkably, we are now able to use this new conformation to hypothesize mechanisms of formation of clustered protocadherin

both in solution and on membranes. We have added these new observations to the Results and Discussion.

The purpose of the experiment was to test whether manual picking from denoised micrographs then training on non-denoised micrographs (compared to manually picking from non-denoised micrographs) helps with 1) Picking more particles of different views, and 2) Picking more particles overall.

We have updated the Results and Discussion sections and the corresponding Figure to show that we have picked more than twice the number of particles and increased the orientational distribution of particles, which as a consequence allowed us to discover a never-before-seen conformation of the clustered protocadherin.

(8) There is also a lack of numerical performance characterization of the algorithm with respect to varying SNRs in the data. To what SNR the algorithm will equally well perform? How will the background look like with even lower SNR after denoising? This can be readily done with simulated noisy micrography with pre-set SNRs, so that the blind test can be compared to the ground truth.

Respectfully, our model improves across a full range of raw micrograph SNRs. In Table 1, we report the raw SNR of each dataset which varies from <0.005 up to about 0.4 along with SNRs after denoising. In addition, Supplemental Tables 1 & 2 report complete quantitative comparisons between denoising methods for micrographs and tomograms. Furthermore, Figure 3a illustrates SNR after denoising for micrographs as the collected dose increases, Figure 3b and Supplemental Figures 13-15 show the visual effects of these titration experiments, and Supplemental Figure 17 shows its downstream effects. Increasing collection dose increases the raw SNR of the micrograph. Here, we see that our denoising method improves SNR over the baseline method at almost every dose of every dataset. To clarify this relationship, we now include an additional plot of raw SNR vs. denoised SNR to accompany Figure 3a (this plot is Supplemental Figure 16).

The Topaz-Denoise models are trained on real data; i.e. micrographs created through the specific image formation models of the microscopes + cameras used across a multitude of real samples. These physical image formation models of the instruments and experimental parameters create the signal and noise in the micrographs. The parameters learned for the noise in the Topaz-Denoise models are based on this real noise formation, not on any simulations. This means that

the model works best on denoising real data collected on similar microscopes and cameras. There is no cryoEM micrograph simulation software that properly mimics real image formation models and so it is not clear what we could learn from denoising those simulated micrographs. We have added the following text to the Introduction to clarify this: “We chose to measure SNR using real cryoEM micrographs because the denoising models were trained on real micrographs generated under real-world conditions that no software accurately simulates.”

(9) One suggestion for authors to consider: besides particle selection and verification, a very helpful use of the denoising tools is high-throughput sample screening, at which step researchers look at the fewer raw micrographs and decide if those are good for scale-up data collection. Lower contrast sometimes can confuse researchers in their judgement about the goodness of the screen sample. With a more reliable denoising tool, the screening could be made more efficiently. The authors could capitalize on this usage for enhancing the significance of the work.

Thank you for the suggestion. We agree that improving the interpretability of micrographs could be very helpful for screening which we now include in the Discussion along with examples of denoising micrographs collected during screening.

(10) The last but not the least is the originality and advancement of this present study seems to be rather limited and incremental. The authors seem to make the least changes to U-Net that was the work of someone else, but merely applied it to the cryo-EM data denoising. While this is certainly a welcoming addition to the present portfolio of software tools in cryo-EM data analysis, the step it meant to tackle might be too small to appeal for its significance in justifying its publication in a journal like Nature Communication. I am all ears to hear any additional evidence to justify its importance.

We thank the reviewer for allowing us to further emphasize the significance of this work. The primary importance of this study is that we provide and thoroughly analyze general models for denoising that are immediately useful to the reader. Not only are these the first (and only) publicly available general denoising models, but we also present the first comprehensive, quantitative evaluation of neural denoising approaches in cryoEM and cryoET. Denoising has the potential to impact many stages of the cryoEM and cryoET analysis process. Topaz-Denoise enables better identification and picking of difficult views of clustered protocadherin, which in our updated manuscript led to finding two conformational states in the data and over twice as many particles as without

denoising. This demonstrates that denoising can allow researchers to find more representative views and more particles overall. Furthermore, denoising can enable shorter exposure imaging which can increase high-resolution and screening cryoEM throughput, and reduce radiation damage to imaged particles. Not only can these reduced exposure micrographs still be used for reconstruction, but the final structure resolution was actually improved. In the revised manuscript, we have also added and analyzed a general 3D denoising model for cryoET. The field of cryoET overlaps considerably with the field of cell biology. CryoET of cells and other cluttered environments suffer significantly from low SNR and cell biologists are typically not adjusted to such a low-SNR regime. Our 3D denoising model lowers the bar for new researchers in both cryoET and cryoEM, thus bridging the gap between the two fields and making our work of broader interest. We have emphasized these points in the Introduction and Discussion.

We think that these results and the widespread interest in deep denoising methods in cryoEM and cryoET warrant presentation in a high impact journal such as *Nature Communications* where the work will reach more readers.

In summary, the authors borrowed an existing CNN architecture called U-Net developed by others previously and applied it to denoise cryo-EM data. Although this is an interesting avenue, the presented study seems to be incomplete, somewhat premature to be published, and lack of numerical performance characterization with differential SNRs to justify its advantage, novelty and significance. However, if the authors can fully address the above concerns, I am glad to take a fresh look at the manuscript.

We present the first general-purpose pre-trained neural denoising models for cryoEM and cryoET as well as the first comprehensive, quantitative comparison of denoising methods in this domain. As clarified above, these methods have widespread ramifications for the EM field and adjacent fields as they can enable new data analysis and process improvements. Although our work builds upon techniques from another field (machine learning), we would like to emphasize that bringing techniques from convolutional neural nets, and in particular U-Net, to cryoEM is a worthy conceptual advance; many top computational biology

publications leverage existing methods to solve novel problems. We believe we have fully addressed the reviewer's concerns as discussed above.

Reviewer #2:

Remarks to the Author:

The authors present an interesting application of a denoising algorithm to cryoEM micrographs. The tool is useful as shown by the authors in their various examples. However, the algorithm is almost trivial given the state of the knowledge at this moment. We recommend the publication of the software as an application note in a journal closer to its final users.

We thank the reviewer for their consideration and appreciation of the usefulness of our method. Although this study is at heart an application of methods from another field (machine learning), we see value in that beyond an application note in a less prominent journal (please see last response to Review 1 just above.) Moreover, we think the enormous interest in deep denoising methods in the cryoEM community and the potential impact of these methods to the cryoEM and cryoET analysis pipelines warrants publication in a high impact journal where it will reach a broad readership. The primary advantage of this work and the provided software are the pre-trained, cryoEM/ET specific denoising models which we characterize thoroughly, apply to a difficult case resulting in a novel structure, evaluate on over a dozen samples, and which the reader may immediately begin using in their cryoEM/ET pipelines. Furthermore, these are the first general-purpose, pre-trained models available for these tasks, development of which required significant data processing and model development independent of the underlying approach.

Reviewer #3:

Remarks to the Author:

This paper introduces and evaluates a method for cryo-EM micrograph denoising based on deep learning and the recent Noise2Noise learning framework. Specifically, the paper identifies the applicability of the Noise2Noise framework based upon the observation that movie frames in cryo-EM can be approximately viewed as noisy samples of the

same underlying "clean" signal. This framework is then used with a variety of network architecture choices and some variations of the specific training loss used.

The proposed denoising algorithm is evaluated for the purposes of visualization and manual particle picking. Quantitative comparisons based on a variation of SNR are also provided. Both quantitative and qualitative results show that the method generally performs well, improving SNR in almost all cases.

The potential of the method is shown by manual selection of particle sets from the denoised micrographs which are then used to train a further automatic particle picker. The results on a challenging dataset (clustered protocadherin) show that this can help in the selection of different particle views for some structures, leading to improvements in resolution and map quality.

Overall it is a solid piece of work and likely to be of interest to cryo-EM practitioners.

We thank the reviewer for appreciating advancements of our study! There are additional advancements of Topaz-Denoise which we would like to emphasize to the reviewer and reader, including faster screening, short exposure, and discerning background contamination from noise. We have now emphasized these points in the Introduction.

Issue: Appropriate Uses of Denoising

=====

I think it's important to note that denoising in cryo-EM can be misused in potentially dangerous ways. The use of denoising as a tool for visualization, exploration and annotation is generally safe and even an ideal application. However, I would very strongly suggest that denoising never be used as part of a reconstruction process. Specifically, denoised micrographs should never be used to perform a reconstruction or refinement. This is because of the increased risk of hallucinated structures and misleading results. It is generally possible that learning-based denoising methods (like the one proposed here) can hallucinate information in the denoised images for a variety of reasons and it is generally not possible to detect. In the context of reconstruction this can lead to a very subtle and hard to detect form of the "Einstein from Noise" phenomena.

We thank the reviewer for bringing this concern to us so that we can bring it to the reader. We agree that a general issue of neural network denoising is a risk of hallucinating features into the augmented data. We have added this important point to a new cautionary paragraph in the Discussion (second-to-last

paragraph). This accompanies supplemental results indicating that denoised particles should not be used for reconstruction (Supplemental Figure 11).

Further, most reconstruction/refinement methods are developed assuming that inputs exhibit the noise which is typically present in cryoEM micrographs and are very likely to fail in both obvious and subtle ways with denoised particle images as inputs. This was, indeed, noted by the authors when they observed that cryoSPARC struggled to do a reconstruction on the denoised particles as shown in Fig 15.

We agree with the reviewer. This is an issue that may present in many complex ways. The alignments and reconstruction in Supplemental Figure 11 possibly exemplify this issue and thus are now presented as a cautionary point in the manuscript. We have added this point to the above-mentioned cautionary paragraph in the Discussion.

I would strongly suggest that a discussion of when and where denoising is appropriate or inappropriate be added to the manuscript as the dangers may not be obvious to many practitioners. In addition, discussion of Supp Figure 15 should be focused on emphasizing why denoised images shouldn't be used for reconstruction.

Thank you. We have added a cautionary paragraph to the Discussion to address these valid points.

Smaller Issues

=====

- References 6-8 are an odd collection of denoising papers. I would suggest just citing 6 as a review article of classical methods, finding a more recent reference for learning based methods and dropping 7,8.

We have removed references 7 & 8.

- Relatedly, denoising has only recently become of strong interest in the ML community. Previously it was primarily of interest in signal processing and computer vision communities so the statement in the manuscript should be correct.

We agree that the rise of deep learning and convolutional neural networks has driven a surge in interest in denoising from an ML perspective. We now draw a clear distinction between CV, signal processing, and ML..

- I would recommend adding a small figure with the specific network architecture used to make it clearer.

We have added a figure of the network architecture as Supplemental Figure 18.

- When training do you only use $\|f(x_a) - x_b\|$ in the loss or do you also use $\|f(x_b) - x_a\|$? I imagine you use both but it's not clear from the text and it seems to imply that only one is used. Ideally both would be used.

We use both. This is now clarified in the text.

- When augmenting the data, how is rotation done? The interpolation that is used during rotations can sometimes dampen noise. Was there any special care taken in this step or is it a simple bilinear interpolation operation in image space?

This is an important detail, thank you for pointing it out. We actually only rotate images by 90, 180, or 270 degrees in order to avoid interpolation. We have added this detail to the text.

- During inference for large images, how are the overlapping regions combined?

We thank the reviewer for allowing us to clarify. Patch denoising includes an option for padding each patch. Topaz-Denoise then extracts the patch size plus the padding on all sides from the original image and denoises the extracted image. The denoised extracted images are then pieced back together. As long as the requested padding is larger than the kernel used for denoising, then the denoised padded pixels that overlap have the exact value of the denoised patch pixels in the adjacent patch. Thus the stitching of regions is seamless. We have added this description to the Methods.

- There are some missing details in the SNR calculation. It also seems like a somewhat strange way to compute the SNR and I think it needs further motivation/justification. I believe in the paper, the magnitude of the "signal" in this case is being defined as the squared difference between the mean value in a signal region and the mean value in a background region. This is likely fine for comparing methods but doesn't really give a true sense of the "SNR" of the denoised image in the sense that the metric is commonly used in image processing.

This approach offers a method for estimating SNR that does not require complete ground truth signal and has been used in other domains before (Hader *et al.* for example), but we agree that these are estimates for purposes of comparing methods and should not be taken as the "true" SNR. We have added the following sentence to the SNR Results section: "Due to the lack of ground

truth in cryoEM, SNR calculations are estimates” to clarify. We have also added a second SNR calculation derived from Frank & Al-Ali for cases where we have full movie frames which offers a better estimate of the true SNR. As a side note, we now also draw a theoretical connection between the SNR of Frank & Al-Ali and the Noise2Noise training framework in the discussion.

- In Supp Figure 14, both plots should have the same range for the Y axis to make them more comparable. This will also make it clearer that the larger particle set was helpful.

Thank you for the suggestion. We updated the figure so that the y-axis for both plots have the same range. We have also binned the maps in Fourier space by 2 before 3DFSC calculation because the resolution is far from Nyquist. This allows for more bins in the histograms to be represented.

Reviewer #4:

Remarks to the Author:

The authors adapted a generalized algorithm for removing unwanted features from images, Noise2Noise, to denoise cryo-electron micrographs, calling it "Topaz-Denoise". It uses the current technology of movies or dose-fractionation to construct two versions of each micrograph. A convolutional neural network is then trained on a subset of the data. The signal is then taken as the common part and the noise the difference. To calculate the enhancement in contrast, a signal-to-noise ratio is calculated as the difference in averages between pairs of particles and background regions, divided by the variance (of the signal or background?).

Thank you for catching the missing subscript in the SNR equation. We divide by the background variance. This has been corrected in the text.

It is an interesting application to enhance the contrast, and is of general interest in image processing. My impression is that it will be mostly useful in picking particles in cryoEM, and may change the way we approach the task. However, care should be taken to emphasize that the denoised particles are not suitable for further processing to generate 3D reconstructions.

We agree with the reviewer that readers should be properly guided on where denoising may be suitable for inclusion in their cryoEM workflow. As mentioned in the response to Reviewer 3, we have added a cautionary paragraph in the

Discussion describing the potential issues of using denoising for cryoEM processing downstream from particle picking.

One general concern is that the SNR is oddly defined (page 10). The typical definition is the ratio of foreground and background variances. Why was a different definition chosen? I would encourage the authors to adopt a definition for the SNR more in line with that commonly used in cryoEM.

This SNR calculation was chosen to average out noise in the foreground calculation. Because we choose very small regions, we assume that the signal is constant and average to remove noise variance. We now clarify this point in the text. Otherwise, this is a typical SNR calculation. We now include a second SNR estimation method, from Frank & Al-Ali, as an alternative to the foreground/background-based estimation in Results section 3.

Page 2:

"Trained on thousands of micrographs across a variety of imaging conditions, these models provide robust denoising without the need to train on a dataset-by-dataset basis."

Does this mean that the trained coefficients from this study could be used in any other case? This would avoid the need for further training.

We thank the reviewer for allowing us to further clarify the main purpose of the manuscript. Yes, the pre-trained models presented in the manuscript and made available in Topaz can be used generally without any further training. We now make this clear throughout the text. We have also clarified the equivalence between "pre-trained" and "general" used throughout the text in the Introduction, "... these general models (or pre-trained models) provide robust denoising without the need to train on a dataset-by-dataset basis."

Page 4:

"Furthermore, we find that the training dataset is important. Intriguingly, the affine, FCNN, and U-net (small) models all perform better than the full U-net model when trained on the small dataset and perform better than their equivalents trained on the large dataset."

I don't understand what the "equivalents" are. Please clarify.

Equivalents are the same model architectures trained on the other dataset. We have updated the text to clarify.

Page 5:

"CryoSparc ab-initio reconstruction using a minimal set of denoised particles is less reliable than using the same set of raw particles (Supplemental Figur 15). Four or five of the six ab-initio reconstructions using the raw particles resulted in the correct overall structure, while only one of the six ab-initio reconstructions using the denoised particles resulted in the correct overall structure"

This is an interesting observation that should feature more prominently in the discussion. I'm not surprised that the denoised particles fared worse. The denoising inevitably removes particle features together with the noise (as does low-pass filtering). It should be more clearly emphasized that the denoised particles should not be used for alignment and reconstruction. It would also be interesting to understand how the denoising changes the spectral characteristics as shown in Supplemental Figure 9.

As requested by Reviewers 3 and 4, we have added a cautionary paragraph to the Discussion to properly advise readers of the limits of denoising in their cryoEM workflow.

Page 7:

The discussion is very short and only covers generalities. This would be the appropriate place to discuss what happens during the denoising procedure and why it enhances the SNR. Also, what happens with other features such as crystalline ice and carbon edges, and how are these handled?

We agree and have substantially expanded the Discussion section. We have added a discussion on denoising's relation to SNR, a conceptual understanding of the network architecture, and a practical understanding of how denoising might affect common features in single particle micrographs.

Moreover, we thank the reviewer for prodding us to investigate the relation between SNR and the objective function we use in Noise2Noise. We found that the concept used in Noise2Noise is related to how SNR has been described since at least in Frank & Al-Ali, 1975, the inception of the field of cryoEM. We describe this relationship explicitly in the Discussion.

Page 10:

In the equation for the SNR I presume v_i refers to the background variance for region i . Please clarify.

Thank you for catching this. This has now been corrected in the text.

REVIEWER COMMENTS

Reviewer #1 (Remarks to the Author):

I appreciate that the authors make numerous revisions in response to my previous comments and that the manuscript has been apparently improved. However, I still have reservations on a few issues and in one case, disagree with the authors' claim and judgement.

(1) My major reservation is about the new claim made by the authors about resolving two structural states of the protocadherin dimer. This conclusion is referred to the data presented in Figure 2b. Although there are some differences in high frequency components of the reconstruction, these differences are way too much beyond the measured resolution (shown in Supplemental Figure 10), which means that the differences are mostly due to random noise or reconstruction error. For all the structural features lower than ~ 12 Å as measured by the authors, I cannot see any reliable, justifiable difference between the acclaimed two states. Beyond subjectively concluding two states (mostly based on errors), the authors offer no validation or any biochemical evidence that can support their assertion of differentiating two conformational states. At this level of resolution, it is very unlikely that any other data or stereochemistry can help with validating their conclusion. I want to remind the authors that the cryo-EM area has long left "blobology" behind. Any demonstration of solving new structural states must be shown at near-atomic resolution, where straightforward stereochemistry validation can be done to ensure the new states are real but not errors. Any efforts that do not meet this "bottom-line" would raise strong doubts and un-certainty about the applicability of the proposed methods.

(2) Are the maps shown in Figure 2b low-pass filtered at the measured resolution? How did the authors validate their 12-Å reconstructions?

(3) While the authors made several welcoming corrections/rectifications to the manuscript, the above issue compromises the importance and potential advantage of the proposed methods. I agree that this could be a useful development to an unknown extent after further correction. However, it remains elusive to this reviewer whether this work belongs to Nature Communications. It seems more suitable for a specialized journal.

Reviewer #3 (Remarks to the Author):

The authors have, in my view, addressed the issues raised by the reviewers in the previous round and significantly improved the paper. At this point I have no further comments to add and would recommend publication. I believe this work will provide a useful tool for the field of cryo-EM and cryo-ET.

Reviewer #4 (Remarks to the Author):

The authors have done an excellent job of improving the manuscript and addressing my comments. I like their addition of denoising tomograms. It seems like a much faster way of doing it than the the best alternative based non-linear anisotropic diffusion. I'm curious how this would compare. The one aspect that may need future attention is the propensity to create artifacts (hallucinations).

Bernard Heymann

Response to Reviewers 2 (Bepler, et al.)

We would like to thank the reviewers for their supportive analysis and comments, which have further improved the manuscript. We have addressed the final reviewer's comments below.

Point-by-point responses (original comments in black; responses in blue).

Reviewer #1:

Remarks to the Author:

I appreciate that the authors make numerous revisions in response to my previous comments and that the manuscript has been apparently improved.

We thank the reviewer for their time and understand their concerns regarding the claims about multiple protocadherin states. We now clarify here and in the manuscript that these conformational states are hypothesized and that this is only one piece of evidence to suggest their existence. The existence or non-existence of multiple states is not the central claim of Results section 2 in our manuscript. The central claim of this section is that Topaz-Denoise allows for particles to be visualized better and thus be picked more completely, which is of general interest to the whole cryoEM field. There has been other work done on clustered protocadherin protein complexes demonstrating significant heterogeneity and flexibility *in situ* on liposomes, and biological follow up work is needed to show the importance/existence of our putative states in purified form.

However, I still have reservations on a few issues and in one case, disagree with the authors' claim and judgement.

(1) My major reservation is about the new claim made by the authors about resolving two structural states of the protocadherin dimer. This conclusion is referred to the data presented in Figure 2b. Although there are some differences in high frequency components of the reconstruction, these differences are way too much beyond the measured resolution (shown in Supplemental Figure 10), which means that the differences are mostly due to random noise or reconstruction error. For all the structural features lower than ~12 Å as measured by the authors, I cannot see any reliable, justifiable difference between the acclaimed two states. Beyond subjectively concluding two states (mostly based on errors), the authors offer no validation or any biochemical evidence that can support their assertion of differentiating two conformational states. At

this level of resolution, it is very unlikely that any other data or stereochemistry can help with validating their conclusion. I want to remind the authors that the cryo-EM area has long left “blobology” behind. Any demonstration of solving new structural states must be shown at near-atomic resolution, where straightforward stereochemistry validation can be done to ensure the new states are real but not errors. Any efforts that do not meet this “bottom-line” would raise strong doubts and un-certainty about the applicability of the proposed methods.

We agree with the reviewer that high resolution structures are needed to verify fine grained structural states. We want to emphasize that our findings constitute a significant advance over the previous highest resolution cryoEM structure of protocadherin - 12 Å vs. 35 Å (see Figure 2a of Brasch *et al.* Nature 2019; doi: 10.1038/s41586-019-1089-3). Furthermore, the structural states we describe here are measurable conformational changes in the twist of the protein. These changes are larger than the 12 Å resolution of the structures we find, which we now illustrate in our revised Figure 2b. We want to emphasize that this is only one piece of evidence to suggest that there are multiple structural states and we now clarify in the Results and Discussion that this is a hypothesis for which we are presenting one small supporting piece of evidence. Beyond this, significant work has been done on the assembly of these proteins on liposomes *in situ* which shows that there is significant flexibility in these complexes (see Figure 3c of Brasch *et al.* Nature 2019; the associated tomogram is available publicly: EMD-9199), so we are far from the first to suggest the existence of multiple states. Indeed, the protein is in effect a cell adhesion molecule that allows neurons to interact with other neurons, which requires substantial flexibility *in vivo*. That said, additional work is certainly needed to verify and describe these states, but that verification is well outside the scope of this work. Our main contribution in this regard is to point out that by picking more particles, enabled by Topaz-Denoise, we are able to substantially improve the resolution of this protein complex in single particle cryoEM. We now clarify these points in the Discussion. As an additional *in silico* verification, we now report the results of multiple replicates of the ab-initio reconstruction procedure all of which discover the same two conformations. We present this now in Supplemental Figure 10.

(2) Are the maps shown in Figure 2b low-pass filtered at the measured resolution? How did the authors validate their 12-Å reconstructions?

The maps in Figure 2b were downsampled in Fourier space by 2 to a pixelsize of 3.68 Å. This has been clarified in the Methods. Refinement was performed in

CryoSPARC using “gold-standard” FSC. This has been clarified in the Methods. As described in the above response, we have reproduced both the open and closed state results 6 times each. The resulting 12 Å reconstruction is clearly better than the previous 35 Å resolution cryoET map shown in Brasch *et al.* Nature 2019 which we now mention.

(3) While the authors made several welcoming corrections/rectifications to the manuscript, the above issue compromises the importance and potential advantage of the proposed methods. I agree that this could be a useful development to an unknown extent after further correction. However, it remains elusive to this reviewer whether this work belongs to Nature Communications. It seems more suitable for a specialized journal.

We now clarify throughout the manuscript that these two states are suggested by this data, but that this is certainly not conclusive proof that these states exist in single particle conditions. Furthermore, we also wish to clarify that these proteins are known to form heterogenous zippered complexes in native conditions, so these findings are only encouraging in that we can potentially discover multiple conformations in single particle cryoEM enabled by improved particle picking with Topaz-Denoise. The claim of one- vs. two-states is not the central finding in Results section 2. The central finding in this section is that by using Topaz-Denoise, researchers can pick more particles no matter how difficult the particle views are to visualize. The clustered protocadherin dataset is an example of a conventionally difficult particle to pick, and Topaz-Denoise allows for more than twice as many particles to be picked, which is a significant result in the cryoEM field. We have emphasized the purpose of this section in the Abstract, Introduction, Results, and Discussion.

We believe that the broad applicability of the pre-trained cryoEM and cryoET denoising models in Topaz-Denoise is of widespread general interest and is functionally useful to all cryoEM and cryoET researchers, thus warranting publication in a widely read journal.

Reviewer #3:

Remarks to the Author:

The authors have, in my view, addressed the issues raised by the reviewers in the previous round and significantly improved the paper. At this point I have no further

comments to add and would recommend publication. I believe this work will provide a useful tool for the field of cryo-EM and cryo-ET.

We thank the reviewer for their positive assessment of our revised manuscript and we agree that Topaz-Denoise pre-trained models are useful to the cyro-EM and cryo-ET communities.

Reviewer #4:

Remarks to the Author:

The authors have done an excellent job of improving the manuscript and addressing my comments. I like their addition of denoising tomograms. It seems like a much faster way of doing it than the the best alternative based non-linear anisotropic diffusion. I'm curious how this would compare. The one aspect that may need future attention is the propensity to create artifacts (hallucinations).

Bernard Heymann

We thank Dr. Heymann for his positive assessment of our revised manuscript.